# Robust Algorithms on Adaptive Inputs from Bounded Adversaries

**Yeshwanth Cherapanamjeri**
UC Berkeley
yeshwanth@berkeley.edu

**Sandeep Silwal**
MIT
silwal@mit.edu

**David P. Woodruff**
Carnegie Mellon University
dwoodruf@andrew.cmu.edu

**Fred Zhang**
UC Berkeley
z0@berkeley.edu

**Qiuyi (Richard) Zhang**
Google Research
qiuyiz@google.com

**Samson Zhou**
UC Berkeley and Rice University
samsonzhou@gmail.com

## Abstract

We study dynamic algorithms robust to adaptive input generated from sources with bounded capabilities, such as sparsity or limited interaction. For example, we consider robust linear algebraic algorithms when the updates to the input are sparse but given by an adversary with access to a query oracle. We also study robust algorithms in the standard centralized setting, where an adversary queries an algorithm in an adaptive manner, but the number of interactions between the adversary and the algorithm is bounded. We first recall a unified framework of (Hassidim et al., 2020; Beimel et al., 2022; Attias et al., 2023) for answering $Q$ adaptive queries that incurs $\widetilde{\mathcal{O}}(\sqrt{Q})$ overhead in space, which is roughly a quadratic improvement over the naïve implementation, and only incurs a logarithmic overhead in query time. Although the general framework has diverse applications in machine learning and data science, such as adaptive distance estimation, kernel density estimation, linear regression, range queries, point queries, and serves as a preliminary benchmark, we demonstrate even better algorithmic improvements for (1) reducing the pre-processing time for adaptive distance estimation and (2) permitting an unlimited number of adaptive queries for kernel density estimation. Finally, we complement our theoretical results with additional empirical evaluations.

## 1 Introduction

Robustness to adaptive inputs or adversarial attacks has recently emerged as an important desirable characteristic for algorithm design. An adversarial input can be created using knowledge of the model to induce incorrect outputs on widely used models, such as neural networks (Biggio et al., 2013; Szegedy et al., 2014; Goodfellow et al., 2015; Carlini & Wagner, 2017; Madry et al., 2018). Adversarial attacks against machine learning algorithms in practice have also been documented in applications such as network monitoring (Chandola et al., 2009), strategic classification (Hardt et al., 2016), and autonomous navigation (Papernot et al., 2016; Liu et al., 2017; Papernot et al., 2017). The need for sound theoretical understanding of adversarial robustness is also salient in situations where successive inputs to an algorithm can be possibly correlated; even if the input is not adversarially generated, a user may need to repeatedly interact with a mechanism in a way such that future updates may depend on the outcomes of previous interactions (Mironov et al., 2011; Gilbert et al., 2012; Bogunovic et al., 2017; Naor & Yogev, 2019; Avdiukhin et al., 2019). Motivated by both practical needs and a lack of theoretical understanding, there has been a recent flurry of theoretical studies of adversarial robustness. The streaming model of computation has especially received significant attention Ben-Eliezer et al. (2021); Hassidim et al. (2020); Woodruff & Zhou (2021); Kaplan et al. (2021); Braverman et al. (2021); Chakrabarti et al. (2022); Ajtai et al. (2022); Chakrabarti et al. (2022); Ben-Eliezer et al. (2022); Assadi et al. (2022); Attias et al. (2023); Dinur et al. (2023); Woodruff et al. (2023). More recently, there have also been a few initial results for dynamic algorithms on adaptive inputs for graph algorithms (Wajc, 2020; Beimel et al., 2021; Bernstein et al., 2022). These works explored the capabilities and limits of algorithms for adversaries that were freely able to choose the input based on previous outputs by the algorithm.

However, in many realistic settings, an adversary is limited in its abilities. For example, adversarial attacks in machine learning are often permitted to only alter the "true" input by a small amount bounded in norm. For the $L_0$ norm, this restriction means that the adversary can only add a sparse noise to the true input. More generally, it seems reasonable to assume that adversarial input is generated from a source that has bounded computation time or bounded interactions with an honest algorithm.

## 1.1 OUR CONTRIBUTIONS

In this paper, we study algorithms robust to adaptive/adversarial input generated from sources with bounded capabilities. We first study dynamic algorithms for adaptive inputs from a source that is restricted in sparsity. Namely, we consider robust linear algebraic algorithms when the updates to the label can be adversarial but are restricted in sparsity. We then study robust algorithms in the standard centralized setting, where an adversary queries an algorithm in an adaptive manner, but the number of interactions between the adversary and the algorithm is bounded. We first show that combining novel subroutines for each of these problems in conjunction with a simple but elegant idea of using differential privacy to hide the internal randomness of various subroutines previously used by Hassidim et al. (2020); Beimel et al. (2022); Attias et al. (2023) suffices to achieve robust algorithms across these different settings.

**Dynamic algorithms on adaptive input for regression.** Motivated by the problem of label shift in machine learning, we consider a dynamic version of least-squares regression, where the labels get updated. In this model, we are given a fixed design matrix and a target label that receives a sequence of updates. After each one, the algorithm is asked to output an estimate of the optimal least-squares objective. The goal of the algorithm is to maintain the objective value within a multiplicative factor $(1 + \varepsilon)$ to the optimal.

More specifically, the algorithm is given a fixed design matrix $\mathbf{A} \in \mathbb{R}^{n \times d}$ with $n \geq d$ and an initial response vector (i.e., label) $\mathbf{b}^{(1)}$, which receives updates over time. We are interested in estimating the least-squares objective value $F(\mathbf{A}, \mathbf{b}) = \min_{\mathbf{x} \in \mathbb{R}^d} \|\mathbf{A}\mathbf{x} - \mathbf{b}\|_2^2$ as the target label $\mathbf{b}$ undergoes updates.

The updates to $\mathbf{b}$ are adaptively chosen by an adversary but can only affect at most $K$ entries of $\mathbf{b}$ per step. Formally, on the $i$-th round:

(1) The adversary provides an update to $K$ entries of $\mathbf{b}^{(i-1)}$, possibly depending on all previous outputs of the algorithm.

(2) The algorithm updates its data structure and outputs an estimate $\widehat{F_i}$ of $F_i = F\left(\mathbf{A}, \mathbf{b}^{(i)}\right)$.

(3) The adversary observes and records the output $\widehat{F_i}$.

The goal of the adversary is to create a sequence of labels $\left(\mathbf{b}^{(i)}\right)_{i=1}^T$ that induces the algorithm to output an inaccurate estimate. To deal with adaptivity, a naïve idea is to treat each step as an independent least-squares regression problem. However, this approach uses a completely new approximation of the objective value for each update, which seems potentially wasteful. On the other hand, any randomness that is shared by computations over multiple updates can potentially be leveraged by the adversary to induce an incorrect output.

Our main result is an algorithm that beats the naïve algorithm in this challenging, adaptively adversarial setting. We provide a general result with run-time dependence on $n, d, K$, and the number of nonzero entries in $\mathbf{A}$, $\mathrm{nnz}(\mathbf{A})$.

**Theorem 1.1** (Informal; see Theorem 2.1). *Let $\kappa(\mathbf{A}) = O(1)$ and $\varepsilon \in (0, 1)$. There exists a dynamic algorithm that given adaptively chosen $K$-sparse updates to $\mathbf{b}$ and a fixed design matrix $\mathbf{A} \in \mathbb{R}^{n \times d}$, outputs a $(1 + \varepsilon)$ approximation to the least-squares objective $F(\mathbf{A}, \mathbf{b}^{(i)})$ every round with high probability. The algorithm uses $\widetilde{\mathcal{O}}\left(\sqrt{K \, \mathrm{nnz}(\mathbf{A})}/\varepsilon^3\right)$ amortized time per step of update.*

Specifically, the update time is $d^{1.5}$ when $K \leq d$ and $n = O(d)$ and square root of the input sparsity when $K = O(1)$. Notice that this significantly betters the naïve approach of treating each step independently and solving for the least-square objective, which requires $\mathrm{nnz}(\mathbf{A}) + \mathrm{poly}(d)$ time by sketching (Woodruff (2014)).

We mention that a recent work by Jiang et al. (2022) considers a row-arrival model for dynamic linear regression. Our setting is different since we allow arbitrary updates to the target label, whereas in their setting the design matrix undertakes incremental change. We note that their algorithm maintains a solution vector, while we focus on the cost only.

**Adaptive query framework.** We then consider robust algorithms in the standard *centralized* setting, where an adversary queries an algorithm in an adaptive manner. In many key algorithmic applications, randomization is necessary to achieve fast query time and efficient storage. This necessitates the need for robust versions of these algorithm which can efficiently employ the power of randomness while also being accurate across multiple possibly correlated inputs. Our main parameters of interest are query time and the space used by a robust algorithm compared to their naïve, non-robust, counterparts.

Formally, we define the model as a two-player game between an algorithm HonestAlg over a data set $X$ and an adversary $\mathcal{A}$ that makes adversarial queries about $X$ to HonestAlg. At the beginning of the game, HonestAlg uses pre-processing time to compute a data structure $\mathcal{D}$ from $X$ to answer future queries from $\mathcal{A}$. The game then proceeds in at most $Q$ rounds for some predetermined $Q$, so that in the $t$-th round, where $t \in [Q]$:

1. $\mathcal{A}$ computes a query $q_t$ on $X$, which depends on all previous responses from HonestAlg.
2. HonestAlg uses $\mathcal{D}$ to output a response $d_t$ to query $q_t$.
3. $\mathcal{A}$ observes and records the response $d_t$.

The goal of $\mathcal{A}$ is to formulate a query $q_t$ for which the algorithm HonestAlg produces an incorrect response $d_t$. We remark that the algorithm may not have access to $X$, after constructing $\mathcal{D}$, to respond to the query $q_t$. On the other hand, $\mathcal{A}$ can use previous outputs to possibly determine the internal randomness of the data structure $\mathcal{D}$ and make future queries accordingly. In this case, the analysis of many randomized algorithms fails because it assumes that the randomness of the algorithm is independent of the input. Consequently, it does not seem evident how to handle $Q$ adaptive queries without implementing $Q$ instances of a non-adaptive data structure, i.e., each instance handles a separate query. Thus, a natural question to ask is whether a space overhead of $\Omega(Q)$ is necessary.

As a preliminary benchmark, we show that a space overhead of $\Omega(Q)$ is unnecessary by giving a unified framework with only an $\widetilde{\mathcal{O}}\left(\sqrt{Q}\right)$ space overhead.

**Theorem 1.2.** *Given a data structure $\mathcal{D}$ that answers a query $q$ with probability at least $\frac{3}{4}$ using space $S$ and query time $T$, there exists a data structure that answers $Q$ adaptive queries, with high probability, i.e., $1 - \frac{1}{\text{poly}(n,Q)}$, using space $\mathcal{O}\left(S\sqrt{Q}\log(nQ)\right)$ and query time $\widetilde{\mathcal{O}}\left(T\log(nQ) + \log^3(nQ)\right)$.*

Theorem 1.2 invokes the framework of Hassidim et al. (2020); Beimel et al. (2022); Attias et al. (2023) to the centralized setting, where a number of queries are made only after the data structure is created.

To concretely instantiate our framework and state an example, we consider the adaptive distance estimation problem defined as follows. In the adaptive distance estimation problem, there exists a set $X = \{\mathbf{x}^{(1)}, \ldots, \mathbf{x}^{(n)}\}$ of $n$ points in $\mathbb{R}^d$. We are also given an accuracy parameter $\varepsilon > 0$. A query is of the form $\mathbf{q}$, and the algorithm must output a $(1 + \varepsilon)$-approximation to $\|\mathbf{x}^{(i)} - \mathbf{q}\|_p$ for all $i$. The trivial solution of storing all $n$ points and computing all $n$ distances to a query point uses space and query time $\mathcal{O}(nd)$. Cherapanamjeri & Nelson (2020) improved the query time to $\widetilde{\mathcal{O}}\left(\frac{n+d}{\varepsilon^2}\right)$ at the cost of using $\widetilde{\mathcal{O}}\left(\frac{(n+d)d}{\varepsilon^2}\right)$ space and $\widetilde{\mathcal{O}}\left(\frac{nd^2}{\varepsilon^2}\right)$ pre-processing time, while permitting an arbitrary number of queries. By comparison, our data structure handles $Q$ queries of approximate distances from a *specified point in $X$*, using query time $\widetilde{\mathcal{O}}\left(\frac{n+d}{\varepsilon^2}\right)$, pre-processing time $\widetilde{\mathcal{O}}\left(\frac{nd\sqrt{Q}}{\varepsilon^2}\right)$, and space $\widetilde{\mathcal{O}}\left(\frac{(n+d)\sqrt{Q}}{\varepsilon^2}\right)$. Thus, in the regime where $d \gg n\sqrt{Q}$, our data structure already improves on the work of Cherapanamjeri & Nelson (2020).

A noticeable weakness of our construction is that the $Q$ queries return only the approximate distance between a query point and a single point in $X$, whereas Cherapanamjeri & Nelson (2020) outputs approximate distances to all points in $X$. Moreover, Cherapanamjeri & Nelson (2022) subsequently improve the pre-processing time to $\widetilde{\mathcal{O}}\left(\frac{nd}{\varepsilon^2}\right)$. Thus we open up our framework to (1) show that it can

be further improved to handle the case where we return the approximate distances of all points in $X$ from $Q$ adaptive query points and (2) achieve pre-processing time $\widetilde{\mathcal{O}}\left(\frac{nd}{\varepsilon^2}\right)$.

**Theorem 1.3.** *There is a data structure which, when instantiated with dataset $X = \{x_i\}_{i \in [n]} \subset \mathbb{R}^d$ and query bound $Q \leq d$, answers any sequence of $Q$ adaptively chosen distance estimation queries correctly with probability at least $0.99$. Furthermore, the space complexity of the data structure is $\widetilde{O}(\varepsilon^{-2} \cdot n\sqrt{Q})$ and the setup and query times are $\widetilde{O}(\varepsilon^{-2} \cdot nd)$ and $\widetilde{O}(\varepsilon^{-2} \cdot (n+d))$, respectively.*

Another application of our framework is the adaptive kernel density estimation problem, where there exists a set $X = \{\mathbf{x}^{(1)}, \ldots, \mathbf{x}^{(n)}\}$ of $n$ points in $\mathbb{R}^d$ and the goal is to output a $(1+\varepsilon)$-approximation to the quantity $\frac{1}{n} \sum_{i \in [n]} k(\mathbf{x}^{(i)}, \mathbf{q})$, for an accuracy parameter $\varepsilon > 0$, a query $\mathbf{q}$, and a kernel function $k$, under the promise that the output is at least some threshold $\tau > 0$. Backurs et al. (2019) give an algorithm for kernel density estimation that uses $\mathcal{O}\left(\frac{1}{\tau \varepsilon^2}\right)$ space and $\mathcal{O}\left(\frac{d}{\sqrt{\tau} \varepsilon^2}\right)$ query time, improving over the standard algorithm that samples $\mathcal{O}\left(\frac{1}{\tau \varepsilon^2}\right)$ points and then uses $\mathcal{O}\left(\frac{d}{\tau \varepsilon^2}\right)$ query time to output the empirical kernel density. However, the analysis for both of these algorithms fails for the adaptive setting, where there can be dependencies between the query and the data structure. By using the data structure of Backurs et al. (2019) as a subroutine, our framework immediately implies an algorithm for adaptive kernel density estimation that uses $\widetilde{\mathcal{O}}\left(\frac{\sqrt{Q}}{\tau \varepsilon^2}\right)$ space and $\mathcal{O}\left(\frac{d \log Q}{\sqrt{\tau} \varepsilon^2}\right)$ query time to answer each of $Q$ adaptive queries. In this case, we are again able to go beyond our framework and give a data structure that handles an unlimited number of adaptive kernel density queries:

**Theorem 1.4.** *Suppose the kernel function $k$ is $L$-Lipschitz in the second variable for some $L > 0$, i.e., $|k(\mathbf{x}, \mathbf{y}) - k(\mathbf{x}, \mathbf{z})| \leq L \|\mathbf{y} - \mathbf{z}\|_2$ for all $\mathbf{x}, \mathbf{y}, \mathbf{z} \in \mathbb{R}^d$. Moreover, suppose that for all $\|\mathbf{x} - \mathbf{y}\|_2 \leq \rho$, we have $k(\mathbf{x}, \mathbf{y}) \leq \frac{\tau}{3}$. Then an algorithm that produces a kernel density estimation data structure $D$ that is $L$-Lipschitz over a set $X$ of points with diameter at most $\Delta$ and outputs a $(1+\varepsilon)$-approximation to KDE queries with value at least $\tau$ with probability at least $1 - \delta$ using space $S(n, \varepsilon, \tau, \log \delta)$ and query time $T(n, \varepsilon, \tau, \log \delta)$, then there exists a KDE data structure that with probability at least $0.99$, outputs a $(1 + \varepsilon)$-approximation to any number of KDE queries with value at least $\tau$ using space $S\left(n, \mathcal{O}(\varepsilon), \mathcal{O}(\tau), \mathcal{O}\left(d \log \frac{(\Delta + \rho)L}{\varepsilon \tau}\right)\right)$ and query time $T\left(n, \mathcal{O}(\varepsilon), \mathcal{O}(\tau), \mathcal{O}\left(d \log \frac{(\Delta + \rho)L}{\varepsilon \tau}\right)\right)$.*

Additionally, we show that our framework guarantees adversarial robustness for a number of other important problems such as nearest neighbor search, range queries, point queries, matrix-vector norm queries, and linear regression. Finally, we supplement our theoretical results with a number of empirical evaluations, which are in the appendix.

## 1.2 OUR TECHNIQUES

**Dynamic regression on adaptive inputs.** Our dynamic algorithm for dynamic maintenance of least-squares objective exploits two main ideas. First, standard results in sketching and sampling show that it suffices to solve for the sketched objective of $\min_{\mathbf{x} \in \mathbb{R}^d} \|\mathbf{SAx} - \mathbf{Sb}\|_2^2$, where $\mathbf{S}$ is an $\ell_2$ subspace embedding for $\mathbf{A}$. Here, we exploit several techniques from numerical linear algebra and in particular use leverage score sampling to obtain a subspace embedding $\mathbf{S}$ of $\mathbf{A}$. By standard results in sketching, a $(1 + \varepsilon)$ optimal solution is given by $\mathbf{x}^* = (\mathbf{SA})^\dagger \mathbf{Sb}$. Moreover, since the goal is to output the objective value instead of the solution vector, we may take a Johnson-Lindenstrauss (JL) sketch to further reduce dimensionality and run-time. This allows us to focus on $\|\mathbf{GAx}^* - \mathbf{Gb}\|_2^2$, where $\mathbf{G} \in \mathbb{R}^{O(\log d) \times n}$ is a JL sketch.

As a result, our algorithm dynamically maintains a solution $\mathbf{GA}(\mathbf{SA})^\dagger \mathbf{b}$ in this sketched space. To achieve that, we first explicitly solve $\mathbf{GA}(\mathbf{SA})^\dagger$ in pre-processing. Since $\mathbf{GA}$ has few rows, this reduces to a small number of linear solves and can be computed fast via conjugate gradient-type methods. To handle the updates, we leverage their sparsity to efficiently maintain the solution and show that each round takes roughly $\mathcal{O}(K)$ time. Amortizing the pre-processing with the update costs over all iterations yields our desired run-time.

Finally, we apply techniques from differential privacy to ensure adversarial robustness, by aggregating independent copies of the algorithm via a private median mechanism. Intuitively, the private mechanism hides the internal randomness of the algorithm and therefore prevents the adversary from otherwise choosing a "bad" input based on knowledge of internal parameters.

**Adaptive query framework.**   Our framework maintains $\widetilde{\mathcal{O}}\left(\sqrt{Q}\right)$ instances of the non-adaptive data structure and crucially uses differential privacy (DP) to protect the internal randomness of the data structures. In addition to our previous results for dynamic regression, the technique of using DP to hide randomness has recently been used in the streaming model (Hassidim et al., 2020; Kaplan et al., 2021) and the dynamic model (Beimel et al., 2021). These works elegantly use the advanced composition property of DP to bound the number of simultaneous algorithms that must be used in terms of the number of times the output changes "significantly" over the course of the stream. In the streaming model, the robust algorithms proceed by instantiating many "hidden" copies of a standard randomized algorithm. As the stream arrives, the algorithms are updated and an answer, aggregated using DP, is reported. Crucially, many of these results exploit the fact that the output answer is monotonic in the stream so that there is a known upper bound on the final output. Thus, the reported answers can only increase by a multiplicative factor at most a logarithmic number of times, which is used to bound the initial number of algorithms which are initialized. In our centralized setting, this can be imagined as setting the parameter $Q$. The main parameter of interest in the streaming literature is the space used by the streaming algorithms, whereas we are concerned with both space usage and query times. Furthermore, stream elements are only accessed one at a time and cannot be processed together unless memory is used. In our case, the dataset is given to us upfront and we can pre-process it to construct a data structure towards solving a centralized problem.

The work by Beimel et al. (2021) shares many of these ideas: the authors are concerned with dynamic graph algorithms where an adversary can update the graph in an adaptive fashion. Similar tools such as multiple randomized initialization and aggregated responses using DP are utilized. The main difference is their parameters of interest: the goal of Beimel et al. (2021) is to have a fast *amortized* update time across many queries. This necessitates the need to "throw away" existing algorithms and start with fresh randomness at intermittent points. In contrast, we study a centralized setting where the underlying dataset is not updated but we wish to answer $Q$ adaptive queries on the dataset.

Inspired by these works, our main framework also uses advanced composition to show the sufficiency of maintaining $\widetilde{\mathcal{O}}\left(\sqrt{Q}\right)$ data structures to answer $Q$ adaptive queries in the centralized setting, which gives a rich set of applications. Moreover, to improve the query time of our algorithms, we further invoke the privacy amplification of sampling to show that it suffices to output the private median of a small subset, i.e., a subset of size $\mathcal{O}\left(\log Q\right)$, of these $\widetilde{\mathcal{O}}\left(\sqrt{Q}\right)$ data structures. Thus our framework only incurs a logarithmic overhead in query time and an $\widetilde{\mathcal{O}}\left(\sqrt{Q}\right)$ overhead in space. Surprisingly, our simple framework gives diverse applications for adaptive algorithms on a number of important problems, including estimating matrix-vector norms, adaptive range query search, adaptive nearest neighbor search, and adaptive kernel density estimation, to name a few. These applications are discussed in depth in Section C.

**Adaptive distance estimation.**   To achieve better pre-processing time for adaptive distance estimation, our main technique is to sample groups of rows from a Hadamard transform and argue that an interaction with a separate group should be considered in separate privacy budgets, effectively arguing that outputting $n$ approximate distances to a single adaptive query only uses one unit of privacy budget. By contrast, our black-box framework charges one unit of privacy budget per approximate distance, so that outputting $n$ approximate distances would use $n$ units of privacy budget.

**Adaptive kernel density estimation.**   Theorem 1.4 is based on showing that with constant probability, our data structure is accurate on all possible queries in $\mathbb{R}^d$. In particular, we first show that our data structure is accurate on a sufficiently fine net of points through a standard union bound argument, which incurs the $d$ overhead compared to the space required to handle a single query. We then show that if the algorithm and the kernel function are both Lipschitz, which is true for sampling-based algorithms and a number of standard kernel functions, then accuracy on the net implies accuracy on all possible points in $\mathbb{R}^d$.

## 2   DYNAMIC REGRESSION UNDER LABEL UPDATES

In this section, we consider the dynamic problem of maintaining the cost of the least-squares regression, where the labels receive adaptively chosen updates. Let $\mathbf{A} \in \mathbb{R}^{n \times d}$ be the design matrix and $\mathbf{b} \in \mathbb{R}^n$ be the target label. A classic problem in numerical linear algebra and optimization is to

solve the $\ell_2$ least-squares regression objective

$$F(\mathbf{A}, \mathbf{b}) = \min_{\mathbf{x} \in \mathbb{R}^d} \|\mathbf{Ax} - \mathbf{b}\|_2^2 = \|\mathbf{AA}^\dagger \mathbf{b} - \mathbf{b}\|_2^2. \qquad (2.1)$$

We consider a dynamic version of the problem, where the label receives adaptively chosen updates. We assume that each update can only affect $K$ entries of the label vector. In this setting, we show:

**Theorem 2.1** (Main theorem; dynamic maintenance of regression cost). *Let $\varepsilon \in (0, 1/4)$ be an error parameter and $\mathbf{b}^{(1)}$ be the initial target label. Given $\varepsilon, \mathbf{A}, \mathbf{b}^{(1)}$, a stream of $T$ adaptively chosen, $K$-sparse updates to the label, Algorithm 4 outputs an estimate $\widehat{F}_i$ such that $\widehat{F}_i = (1 \pm \varepsilon)F(\mathbf{A}, \mathbf{b}^{(i)})$ for all $i$ with high probability.*

*Furthermore, the algorithm requires a preprocessing step in time $\widetilde{\mathcal{O}}(\text{nnz}(\mathbf{A}) + \text{poly}(d))$. The amortized update time of the algorithm is*

$$\widetilde{\mathcal{O}}\left(\sqrt{K\,\text{nnz}(\mathbf{A})}\left(\sqrt{\kappa(\mathbf{A})} + \varepsilon^{-3}\right)\right).$$

We defer the technical details to Appendix 3. Here, we describe the main ideas of the algorithm.

At a high-level, our algorithm implements a *sketch-and-solve* strategy. First, in pre-processing, the algorithm samples a leverage score sketching matrix $\mathbf{S} \in \mathbb{R}^{k \times n}$, where $k = \mathcal{O}\left(d \log d/\varepsilon^2\right)$. This provides a $(1 + \varepsilon)$ $\ell_2$ subspace embedding for $\mathbf{A}$. Standard results in sketching imply that it suffices to solve for the sketched objective of $\min_{\mathbf{x} \in \mathbb{R}^d} \|\mathbf{SAx} - \mathbf{Sb}\|_2^2$ (Sarlos (2006); Clarkson & Woodruff (2013; 2017); Woodruff (2014)). Let $\widehat{\mathbf{A}} = \mathbf{SA}$. A $(1 + \varepsilon)$ optimal solution is thus given by $\widehat{\mathbf{A}}^\dagger \mathbf{b}$. Moreover, our goal is to maintain the regression cost, rather than this solution vector. Hence, we can apply the Johnson–Lindenstrauss lemma and focus on

$$\min_{\mathbf{x} \in \mathbb{R}^d} \|\mathbf{SAx} - \mathbf{Sb}\|_2^2 \approx \|\mathbf{GA}(\mathbf{SA})^\dagger \mathbf{Sb} - \mathbf{Gb}\|_2^2, \qquad (2.2)$$

where $\mathbf{G} \in \mathbb{R}^{\mathcal{O}(\log n/\varepsilon^2) \times n}$ is a JL sketch.

To track the cost value dynamically, the algorithm first computes and stores $\mathbf{M} = \mathbf{GA}(\mathbf{SA})^\dagger$ in pre-processing. In the 1-st step, given the initial target label $\mathbf{b}^{(1)}$, the algorithm computes $\mathbf{Sb}^{(i)}$, $\mathbf{M}\left(\mathbf{Sb}^{(1)}\right)$ and $\mathbf{Gb}^{(1)}$. Then it outputs $\widehat{F}_1 = \|\mathbf{MSb}^{(1)} - \mathbf{Gb}^{(1)}\|_2^2$ as an estimate of the regression cost. For the later steps, we show how to maintain $\mathbf{Gb}^{(i)}, \mathbf{Sb}^{(i)}$ efficiently, by exploiting the sparsity of the updates. Finally, to remain robust under adaptive inputs, we aggregate multiple copies using private median and carefully balance the parameters to achieve the run-time guarantee.

## 3  DETAILS ON DYNAMIC REGRESSION

In this section, we consider the dynamic problem of maintaining the cost of the least-squares regression, where the labels receive adaptively chosen updates.

We first introduce the basic setting of the problem in Section 3.1. In Section 3.2, we design a key subroutine under non-adaptive updates. The data structure enjoys a nearly linear update time. This allows us to aggregate multiple copies of the procedure and thereby efficiently ensure adversarial robustness against an adaptive adversary. The argument is via an application of differential privacy and detailed subsequently in Section 3.3.

### 3.1  BASIC SETTING

Let $\mathbf{A} \in \mathbb{R}^{n \times d}$ be the design matrix and $\mathbf{b} \in \mathbb{R}^n$ be the target label. A classic problem in numerical linear algebra and optimization is to solve the $\ell_2$ least-squares regression objective

$$F(\mathbf{A}, \mathbf{b}) = \min_{\mathbf{x} \in \mathbb{R}^d} \|\mathbf{Ax} - \mathbf{b}\|_2^2 = \|\mathbf{AA}^\dagger \mathbf{b} - \mathbf{b}\|_2^2. \qquad (3.1)$$

We consider a dynamic version of the problem, where the design matrix $\mathbf{A}$ remains unchanged. However, at each step (at most) $K$ entries of $\mathbf{b}$ undergo an update. Moreover, we assume that the updates are chosen adaptively by an adversary in the following manner.

- The algorithm starts by receiving the input $\mathbf{A} \in \mathbb{R}^{n \times d}$ and $\mathbf{b}^{(1)} \in \mathbb{R}^n$.

- In the $i$-th step, the algorithm outputs an estimate $\widehat{F}_i$ of the cost $F(\mathbf{A}, \mathbf{b}^{(i)})$, where $\mathbf{b}^{(i)}$ is the target label corresponding to the step.

- The adversary observes $\widehat{F}_i$ and updates at most $K$ labels to form $\mathbf{b}^{(i)}$.

Let $\mathbf{b}^{(1)}, \mathbf{b}^{(2)}, \ldots, \mathbf{b}^{(T)} \in \mathbb{R}^n$ be the resulting sequence of labels over $T$ steps. The goal of the algorithm is to output a $(1 + \varepsilon)$ approximation to the optimal cost at every step, while minimizing the update time.

## 3.2 Dynamic Algorithm for Oblivious Inputs

In this section, we provide a key subroutine that maintains a data structure under oblivious updates. On a high-level, the data structure aims to enable a *sketch-and-solve* strategy dynamically. The main ideas are two fold: (1) apply randomized sketching to reduce dimensionality and therefore the run-time, and (2) exploit the sparsity of the updates to argue that the regression costs can be maintained efficiently.

Before delving into the technical details, we give an overview of the algorithm.

**Overview of the algorithm.** We start by assuming that the algorithm has access to $\mathcal{D}_{LS}$ (via Lemma A.11), the row leverage score sampling data structure for $\mathbf{A}$. In preprocessing, the algorithm samples a leverage score sketching matrix $\mathbf{S} \in \mathbb{R}^{k \times n}$ from $\mathcal{D}_{LS}$, where $k = \mathcal{O}(d \log d / \varepsilon^2)$. This provides a $(1 + \varepsilon)$ $\ell_2$ subspace embedding for $\mathbf{A}$. Standard results in sketching imply that it suffices to solve for the sketched objective of $\min_{\mathbf{x} \in \mathbb{R}^d} \|\mathbf{SAx} - \mathbf{Sb}\|_2^2$ Sarlos (2006); Clarkson & Woodruff (2013; 2017); Woodruff (2014). Let $\widehat{\mathbf{A}} = \mathbf{SA}$. Then a $(1 + \varepsilon)$ optimal solution is thus given by $\widehat{\mathbf{A}}^\dagger \mathbf{b}$. Moreover, our goal is to maintain the regression cost, rather than this solution vector. Hence, we can apply Johnson–Lindenstrauss lemma and focus on

$$\min_{\mathbf{x} \in \mathbb{R}^d} \|\mathbf{SAx} - \mathbf{Sb}\|_2^2 \approx \left\|\mathbf{GA}(\mathbf{SA})^\dagger \mathbf{Sb} - \mathbf{Gb}\right\|_2^2, \tag{3.2}$$

where $\mathbf{G} \in \mathbb{R}^{\mathcal{O}(\log n / \varepsilon^2) \times n}$ is a JL sketch.

Next, we describe how to track the cost value dynamically. We stress that the sketching matrices $\mathbf{S}$ and $\mathbf{G}$ are sampled upfront in the preprocessing stage and remain fixed afterwards. The algorithm stores $\mathbf{G}$ and $\mathbf{M} = \mathbf{GA}(\mathbf{SA})^\dagger$, both computed in preprocessing. Meanwhile, it maintains $\mathbf{Gb}^{(i)}, \mathbf{Sb}^{(i)}$, initialized at $i = 1$. In the first step, given the initial target label $\mathbf{b}^{(1)}$, the algorithm computes $\mathbf{Sb}^{(i)}$, $\mathbf{M}\left(\mathbf{Sb}^{(1)}\right)$ and $\mathbf{Gb}^{(1)}$. Then it outputs $\widehat{F}_1 = \left\|\mathbf{MSb}^{(1)} - \mathbf{Gb}^{(1)}\right\|_2^2$ as an estimate of the regression cost.

Let's consider the $i$-th step, where the label is updated to $\mathbf{b}^{(i)}$. First, we read the $K$ labels that get changed and update $\mathbf{Sb}^{(i-1)}$ to $\mathbf{Sb}^{(i)}$ accordingly. This can be done in $\mathcal{O}(K)$ time. Finally, we simply compute $\mathbf{M}(\mathbf{Sb}^{(i)})$ and $\mathbf{Gb}^{(i)}$ and output $\widehat{F}_i = \left\|\mathbf{MSb}^{(i)} - \mathbf{Gb}^{(i)}\right\|_2^2$. We store $\mathbf{Gb}^{(i)}$ for the next iteration.

We now describe the algorithm formally, followed by an analysis of its run-time and accuracy.

**Formal description of the algorithm.** We assume $\mathcal{D}_{LS}$ for $\mathbf{A}$ is given. The data structure is initialized by drawing the sketching matrices $\mathbf{G}$ and $\mathbf{S}$. We also compute $\mathbf{M} = \mathbf{GSA}(\mathbf{SA})^\dagger$ in preprocessing. This matrix is stored explicitly throughout.

---

**Algorithm 1** Initialize the data structure, i.e., preprocessing

---

**Input:** Design matrix $\mathbf{A} \in \mathbb{R}^{n \times d}$, initial label $\mathbf{b}^{(1)} \in \mathbb{R}^n$, $\mathcal{D}_{LS}, \varepsilon \in (0, 1)$

1: Let $k = \Theta\left(d \log d / \varepsilon^2\right)$
2: Sample a $(1 + \varepsilon/2)$ $\ell_2$ leverage score row sampling matrix $\mathbf{S} \in \mathbb{R}^{k \times n}$ for $\mathbf{A}$ from $\mathcal{D}_{LS}$.
3: Sample a JL sketch matrix $\mathbf{G} \in \mathbb{R}^{C \varepsilon^{-2} \log n \times n}$, for a sufficiently large $C$, by Theorem A.6.
4: Compute and store $\mathbf{M} = \mathbf{GA}(\mathbf{SA})^\dagger$.

---

At each step, the algorithm computes $\mathbf{Sb}^{(i)}$ by reading all $K$ entries of $\mathbf{b}^{i-1}$ that are updated in the step. After that, compute $\mathbf{M}(\mathbf{Sb}^{(i)})$ and $\mathbf{Gb}^{(i)}$ and output $\left\|\mathbf{Mb}^{(i)} - \mathbf{Gb}^{(i)}\right\|_2^2$. The algorithm is formally given by Algorithm 2.

---

**Algorithm 2** Update data structure and maintain regression cost

---

**Input:** Matrices $\mathbf{G} \in \mathbb{R}^{C\varepsilon^{-2}\log n \times n}, \mathbf{S} \in \mathbb{R}^{k \times n}, \mathbf{M} \in \mathbb{R}^{\widetilde{\mathcal{O}}(1/\varepsilon^2) \times k}$ and the label $\mathbf{b}^{(i)}$
**Output:** Estimate of the regression cost $F\left(\mathbf{A}, \mathbf{b}^{(i)}\right)$
  1: Compute $\mathbf{Sb}^{(i)}$ by reading all $K$ entries of $\mathbf{b}^{(i-1)}$ that are updated.
  2: Compute $\mathbf{M}\left(\mathbf{Sb}^{(i)}\right)$ and $\mathbf{Gb}^{(i)}$.    ▷Store $\mathbf{MSb}^{(i)}, \mathbf{Sb}^{(i)}, \mathbf{Gb}^{(i)}$ for the next round.
  3: Output $\widehat{F}_i = \left\|\mathbf{MSb}^{(i)} - \mathbf{Gb}^{(i)}\right\|_2^2$.

---

**Analysis of the algorithm.**    We now analyze the run-time of the algorithm. First, consider the preprocessing stage performed by Algorithm 1.

**Lemma 3.1** (Preprocessing time). *Assuming access to the leverage score sampling data structure $\mathcal{D}_{LS}$, the preprocessing time of Algorithm 1 is*

$$\mathcal{O}\left(\sqrt{\kappa(\mathbf{A})}\,\mathrm{nnz}(\mathbf{A})\log\frac{1}{\varepsilon} + \frac{\mathrm{nnz}(\mathbf{A})}{\varepsilon^2}\log n + \frac{d}{\varepsilon^2}\log n\right). \tag{3.3}$$

*Proof.* By Lemma A.11, the guarantee of the sampling data structure $\mathcal{D}_{LS}$, it takes $\mathcal{O}(k\log(nd))$ time to obtain a leverage score sample $\mathbf{S}$ of size $k$. Drawing the JL sketch is straightforward, and standard constructions such as i.i.d. Gaussian entries require $\mathcal{O}(k\log n/\varepsilon^2)$ times to form $\mathbf{G}$.

Finally, we need to compute $\mathbf{M}$. Computing $\mathbf{GA}$ requires $\mathcal{O}\left(\frac{\mathrm{nnz}(\mathbf{A})}{\varepsilon^2}\log n\right)$ time by sparse matrix multiplication. Moreover, since $\mathbf{GA}$ is a matrix of $\mathcal{O}\left(\frac{\log n}{\varepsilon^2}\right)$ rows, then computing $(\mathbf{GA})(\mathbf{SA})^\dagger$ reduces to $\mathcal{O}\left(\frac{\log n}{\varepsilon^2}\right)$ number of linear system solves with respect to $\mathbf{SA} \in \mathbb{R}^{k \times d}$. By conjugate gradient type methods, since $\kappa(\mathbf{SA}) = (1 \pm \varepsilon)\kappa(\mathbf{A})$, each solve can be achieved to high accuracy in $\mathcal{O}\left(\sqrt{\kappa(\mathbf{A})}\log(1/\varepsilon)\right)$ number of matrix-vector products with respect to $\mathbf{A}$ Golub & Van Loan (2013). In total, this gives a run-time of $\mathcal{O}\left(\sqrt{\kappa(\mathbf{A})}\,\mathrm{nnz}(\mathbf{A})\log(1/\varepsilon)\right)$.    □

**Lemma 3.2** (Update time). *The update time of Algorithm 2 is $\mathcal{O}\left(\frac{K}{\varepsilon^2}\log n\right)$ per step.*

*Proof.* First, the algorithm reads the $K$ entries that are updated and compute the $\mathbf{Sb}^{(i)}$ from $\mathbf{Sb}^{(i-1)}$. This step takes $\mathcal{O}(K)$ time, since we just need to update the entries that lie in the support of the row sampling matrix $\mathbf{S}$. Similarly, in step 2 of Algorithm 2 we can update $\mathbf{Gb}^{(i-1)}$ to $\mathbf{Gb}^{(i)}$ in $\mathcal{O}(K\log n/\varepsilon^2)$ time. Since $\mathbf{S}$ is a row sampling matrix and $\mathbf{b}^{(i)}$ only has $K$ entries updated, then $\mathbf{Sb}^{(i)}$ has at most $K$ entries updated as well. It follows that given $\mathbf{M}\left(\mathbf{Sb}^{(i-1)}\right)$ from the prior round, $\mathbf{M}\left(\mathbf{Sb}^{(i)}\right)$ can be updated in $\mathcal{O}\left(\frac{K}{\varepsilon^2}\log n\right)$ time.    □

**Lemma 3.3** (Accuracy). *Given a stream of $T = \mathcal{O}(d^2)$ non-adaptive updates and error parameter $\varepsilon \in (0, 1/4)$, Algorithm 2 outputs an estimate $\widehat{F}_i$ of the regression cost $F(\mathbf{A}, \mathbf{b}^{(i)})$ such that $\widehat{F}_i = (1 \pm \varepsilon)F(\mathbf{A}, \mathbf{b}^{(i)})$ for all $i$ with high probability.*

*Proof.* First, we apply the subspace embedding property of $\mathbf{S}$. This implies that with high probability,

$$\min_{\mathbf{x}}\left\|\mathbf{SAx} - \mathbf{Sb}^{(i)}\right\|_2^2 = (1 \pm \varepsilon/2)\min_{\mathbf{x}}\left\|\mathbf{Ax} - \mathbf{b}^{(i)}\right\|_2^2.$$

Apply the JL lemma (Theorem A.6), where we consider the collection of $\mathcal{O}\left(d^2\right)$ $(1+\varepsilon)$ optimal predictions $\{\mathbf{y}_i^*\}_{i=1}^T$ with $\mathbf{y}_i^* = \mathbf{A}(\mathbf{SA})^\dagger\mathbf{b}^{(i)}$. Via union bound, we have that with high probability for all $i \in [T]$

$$\left\|\mathbf{Gy}_i^* - \mathbf{Gb}^{(i)}\right\|_2^2 = (1 \pm \varepsilon/2)\left\|\mathbf{y}_i^* - \mathbf{b}^{(i)}\right\|_2^2.$$

Our algorithm precisely solves for $\mathbf{y}_i^*$ each iteration. Combining the two equations above finishes the proof. $\qquad\square$

## 3.3 DYNAMIC ALGORITHM WITH ADVERSARIAL ROBUSTNESS

To put everything together and ensure adversarial robustness, we use a standard approach of Hassidim et al. (2020); Beimel et al. (2022); Attias et al. (2023). Our full algorithm maintains $\Gamma = \mathcal{O}\left(\sqrt{T}\log(nT)\right)$ independent copies of the key subroutine for $T = \mathcal{O}\left(\frac{\mathrm{nnz}(\mathbf{A})}{\varepsilon^2 K}\right)$. Then at each step, we output the private median of the outputs of these copies. Advanced composition of DP ensures robustness up to $T$ rounds. Afterwards, the algorithm reboots by rebuilding the copies, using fresh randomness independently for sampling and computing the sketching matrices.

---

**Algorithm 3** Preprocessing step for Algorithm 4

---

**Input:** A design matrix $\mathbf{A} \in \mathbb{R}^{n \times d}$, an approximation factor $\varepsilon \in (0, 1)$.
**Output:** The leverage score sampling data structure $\mathcal{D}_{LS}$ for $\mathbf{A}$.
 1: Compute the approximate row leverage scores of $\mathbf{A}$. $\qquad\qquad\qquad\qquad\triangleright$Lemma A.11
 2: Build and output the data structure $\mathcal{D}_{LS}$

---

**Algorithm 4** Dynamic algorithm for maintaining regression cost under adaptive updates

---

**Input:** A sequence of target labels $\left\{\mathbf{b}^{(i)}\right\}_{i=1}^m$ and a fixed design matrix $\mathbf{A} \in \mathbb{R}^{n \times d}$, an approximation factor $\varepsilon \in (0, 1)$, the leverage score sampling data structure $\mathcal{D}_{LS}$ for $\mathbf{A}$.
**Output:** Estimates of the regression cost $F(\mathbf{A}, \mathbf{b}^{(i)})$ under adaptively chosen updates to $\mathbf{b}$.
 1: **for** every epoch of $T = \mathcal{O}\left(\frac{\mathrm{nnz}(\mathbf{A})}{\varepsilon^2 K}\right)$ updates **do**
 2: $\qquad$ Initialize $\Gamma = \mathcal{O}\left(\sqrt{T}\log(nT)\right)$ independent instances of the data structure in Section 3.2 via Algorithm 1.
 3: $\qquad$ Run PrivMed on the $\Gamma$ instances with privacy parameter $\varepsilon' = \mathcal{O}\left(\frac{1}{\sqrt{T}\log(nT)}\right)$ with failure probability $\delta = \frac{1}{\mathrm{poly}(m,T)}$.
 4: $\qquad$ For each query, return the output of PrivMed.

---

**Theorem 3.4.** *[Main theorem; dynamic maintenance of regression cost] Let $\varepsilon \in (0, 1/4)$ be an error parameter and $\mathbf{b}^{(1)}$ be the initial target label. Given $\varepsilon, \mathbf{A}, \mathbf{b}^{(1)}$, a stream of $T$ adaptively chosen, $K$-sparse updates to the label, Algorithm 4 outputs an estimate $\widehat{F}_i$ such that $\widehat{F}_i = (1 \pm \varepsilon)F(\mathbf{A}, \mathbf{b}^{(i)})$ for all $i$ with high probability.*

*Furthermore, the algorithm requires a preprocessing step in time $\widetilde{\mathcal{O}}\left(\mathrm{nnz}(\mathbf{A}) + \mathrm{poly}(d)\right)$. The amortized update time of the algorithm is*

$$\widetilde{\mathcal{O}}\left(\sqrt{K\,\mathrm{nnz}(\mathbf{A})}\left(\sqrt{\kappa(\mathbf{A})} + \varepsilon^{-3}\right)\right)$$

*per round.*

We defer the proof of Theorem 3.4 and a discussion on a deterministic algorithm to Section B.

ACKNOWLEDGEMENTS

Sandeep Silwal is supported by an NSF Graduate Research Fellowship under Grant No. 1745302, and NSF TRIPODS program (award DMS-2022448), NSF award CCF-2006798, and Simons Investigator Award (via Piotr Indyk). This work was done in part while David P. Woodruff was at Google Research and supported in part by a Simons Investigator Award and by the National Science Foundation under Grant No. CCF-1815840. Fred Zhang is supported by ONR grant N00014-18-1-2562. This work was done in part while Samson Zhou was at Carnegie Mellon University and supported in part by a Simons Investigator Award and by the National Science Foundation under Grant No. CCF-1815840.

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

## A  PRELIMINARIES

**Notations**   In this paper, we use $[n]$ for a positive integer $n > 0$ to denote the set $\{1, \ldots, n\}$. We use $\text{poly}(n)$ to denote a fixed polynomial in $n$. We say an event occurs with high probability if it occurs with probability $1 - \frac{1}{\text{poly}(n)}$. For real numbers $a, b$ and positive $\varepsilon$, we say $a = (1 \pm \varepsilon)b$ if $(1 - \varepsilon)b \leq a \leq (1 + \varepsilon)b$. Let $\mathbf{e}_i \in \mathbb{R}^n$ be the $i$'th standard basis vector. Let $\mathbf{X}^+$ denote the Moore-Penrose pseudo-inverse of matrix $\mathbf{X}$. Let $\|\mathbf{X}\|$ denote the operator norm of $\mathbf{X}$. Let $\kappa(\mathbf{X}) = \|\mathbf{X}^+\| \|\mathbf{X}\|$ denote the condition number of $\mathbf{X}$.

### A.1  DIFFERENTIAL PRIVACY

Much of our technical results leverage tools from DP. We recall its definition and several key statements.

**Definition A.1** (Differential privacy, Dwork et al. (2006))**.** *Given $\varepsilon > 0$ and $\delta \in (0, 1)$, a randomized algorithm $\mathcal{A} : \mathcal{X}^* \to \mathcal{Y}$ is $(\varepsilon, \delta)$-differentially private if, for every neighboring datasets $S$ and $S'$ and for all $E \subseteq \mathcal{Y}$,*

$$\mathbf{Pr}\left[\mathcal{A}(S) \in E\right] \leq e^\varepsilon \cdot \mathbf{Pr}\left[\mathcal{A}(S') \in E\right] + \delta.$$

**Theorem A.2** (Amplification via sampling, e.g., Bun et al. (2015))**.** *Let $\mathcal{A}$ be an $(\varepsilon, \delta)$-differentially private algorithm for $\varepsilon \leq 1$, $\delta \in (0, 1)$. Given a database $S$ of size $n$, let $\mathcal{A}'$ be the algorithm that constructs a database $T \subset S$ by subsampling (with replacement) $s \leq \frac{n}{2}$ rows of $S$ and outputs $\mathcal{A}(T)$. Then $\mathcal{A}'$ is $(\varepsilon', \delta')$-differentially private for*

$$\varepsilon' = \frac{6\varepsilon k}{n}, \qquad \delta' = \exp(6\varepsilon k/n)\, \frac{4k\delta}{n}.$$

**Theorem A.3** (Private median, e.g., Hassidim et al. (2020))**.** *Given a database $\mathcal{D} \in X^*$, there exists an $(\varepsilon, 0)$-differentially private algorithm* PrivMed *that outputs an element $x \in X$ such that with probability at least $1 - \delta$, there are at least $\frac{|S|}{2} - k$ elements in $S$ that are at least $x$, and at least $\frac{|S|}{2} - k$ elements in $S$ in $S$ that are at most $x$, for $k = \mathcal{O}\left(\frac{1}{\varepsilon} \log \frac{|X|}{\delta}\right)$.*

**Theorem A.4** (Advanced composition, e.g., Dwork et al. (2010))**.** *Let $\varepsilon, \delta' \in (0, 1]$ and let $\delta \in [0, 1]$. Any mechanism that permits $k$ adaptive interactions with mechanisms that preserve $(\varepsilon, \delta)$-differential privacy guarantees $(\varepsilon', k\delta + \delta')$-differential privacy, where $\varepsilon' = \sqrt{2k \ln \frac{1}{\delta'}} \cdot \varepsilon + 2k\varepsilon^2$.*

**Theorem A.5** (Generalization of DP, e.g., Dwork et al. (2015); Bassily et al. (2021))**.** *Let $\varepsilon \in (0, 1/3)$, $\delta \in (0, \varepsilon/4)$, and $n \geq \frac{1}{\varepsilon^2} \log \frac{2\varepsilon}{\delta}$. Suppose $\mathcal{A} : X^n \to 2^X$ is an $(\varepsilon, \delta)$-differentially private algorithm that curates a database of size $n$ and produces a function $h : X \to \{0, 1\}$. Suppose $\mathcal{D}$ is a distribution over $X$ and $S$ is a set of $n$ elements drawn independently and identically distributed from $\mathcal{D}$. Then*

$$\Pr_{S \sim \mathcal{D}, h \leftarrow \mathcal{A}(S)} \left[ \left| \frac{1}{|S|} \sum_{x \in S} h(x) - \mathbb{E}_{x \sim \mathcal{D}}[h(x)] \right| \geq 10\varepsilon \right] < \frac{\delta}{\varepsilon}.$$

## A.2 NUMERICAL LINEAR ALGEBRA

Our results on dynamic regression relies upon some tools in numerical linear algebra. We first recall the dimensionality reduction techniques.

**Theorem A.6** (Johnson-Lindenstrauss transformation, $\varepsilon$-JL)**.** *Given $\varepsilon > 0$, there exists a family of random maps $\Pi_{m,d} \in \mathbb{R}^{m \times d}$ with $m = \mathcal{O}\left(\frac{1}{\varepsilon^2}\right)$ such that for any $\mathbf{x} \in \mathbb{R}^d$, we have*

$$\Pr_{\Pi \sim \Pi_{m,d}} [(1 - \varepsilon)\|\mathbf{x}\|_2 \leq \|\Pi\mathbf{x}\|_2 \leq (1 + \varepsilon)\|\mathbf{x}\|_2] \geq \frac{3}{4}.$$

*Moreover, $\Pi\mathbf{x}$ takes $\mathcal{O}\left(\frac{d}{\varepsilon^2}\right)$ time to compute.*

**Theorem A.7** (Fast JL)**.** *Given $\varepsilon > 0$, there exists a family of random maps $\Pi_{m,d} \in \mathbb{R}^{m \times d}$ with $m = \mathcal{O}\left(\frac{\log d}{\varepsilon^2}\right)$ such that for any $\mathbf{x} \in \mathbb{R}^d$, we have*

$$\Pr_{\Pi \sim \Pi_{m,d}} [(1 - \varepsilon)\|\mathbf{x}\|_2 \leq \|\Pi\mathbf{x}\|_2 \leq (1 + \varepsilon)\|\mathbf{x}\|_2] \geq \frac{3}{4}.$$

*Moreover, $\Pi\mathbf{x}$ takes $\mathcal{O}\left(\frac{\log d}{\varepsilon^2} + d \log d\right)$ time to compute.*

A row sampling matrix $\mathbf{S}$ has rows that are multiples of natural basis vectors, so that $\mathbf{SA}$ is a (weighted) sample of the rows of $\mathbf{A}$. A column sampling matrix is defined similarly. The size of a row/column sampling matrix is defined as the number of rows/columns it samples. The leverage score of the $i$th row $\mathbf{a}_i^\top$ of $\mathbf{A}$ is

$$\tau_i(\mathbf{A}) \stackrel{\text{def}}{=} \mathbf{a}_i^\top \left(\mathbf{A}^\top \mathbf{A}\right)^+ \mathbf{a}_i.$$

For a survey on leverage score and applications, we refer the reader to Mahoney (2011).

**Definition A.8** (Leverage score sampling)**.** *Let $\mathbf{u}$ be a vector of leverage score overestimates, i.e., $\tau_i(\mathbf{A}) \leq \mathbf{u}_i$. Let $\alpha$ be a sampling rate parameter and $c$ be a fixed positive constant. For each row, we define a sampling probability $p_i = \min\{1, \alpha \cdot u_i c \log d\}$. The leverage score sampling matrix is a row sampling matrix $\mathbf{S}$ with independently chosen entries such that $\mathbf{S}_{ii} = \frac{1}{\sqrt{p_i}}$ with probability $p_i$ and $0$ otherwise.*

**Definition A.9** (Subspace embedding)**.** *A $(1 \pm \varepsilon)$ $\ell_2$ subspace embedding for the column space of an $n \times d$ matrix $\mathbf{A}$ is a matrix $\mathbf{S}$ for which for all $\mathbf{x} \in \mathbb{R}^d$*

$$\|\mathbf{SAx}\|_2^2 = (1 \pm \varepsilon)\|\mathbf{Ax}\|_2^2.$$

**Theorem A.10** (Leverage sampling implies subspace embedding, Theorem 17 of Woodruff (2014))**.** *Let $\alpha = \varepsilon^{-2}$ and $c$ be a sufficiently large constant. With high probability, the leverage score sampling matrix is a $(1 \pm \varepsilon)$ $\ell_2$ subspace embedding. Furthermore, it has size $\mathcal{O}\left(d \log d / \varepsilon^2\right)$.*

The approximate leverage scores can be computed in input-sparsity time. Afterwards, repeated sampling from the leverage score distribution can be done efficiently using the binary tree data structure in quantum-inspired numerical linear algebra.

**Lemma A.11** (Leverage score computation and sampling data structure; see Woodruff (2014); Chepurko et al. (2022)). *Let $\mathbf{A} \in \mathbb{R}^{n \times d}$. There exists an algorithm that given $\mathbf{A}$ outputs a vector of row leverage score overestimates with high probability and in run-time $\widetilde{\mathcal{O}}\left(\mathrm{nnz}(\mathbf{A}) + \mathrm{poly}(d)\right)$.*

*Furthermore, there exists a sampling data structure $\mathcal{D}_{LS}$ that stores the row leverage scores of $\mathbf{A}$ such that given a positive integer $m \leq n$, returns a leverage score sample of $\mathbf{A}$ of size $m$ in $\mathcal{O}\left(m \log(mn)\right)$ time. In total, the pre-processing takes $\mathcal{O}\left(\mathrm{nnz}(\mathbf{A}) + \mathrm{poly}(d)\right)$ time.*

# B    ADDITIONAL DETAILS ON DYNAMIC REGRESSION

**Theorem 3.4.** *[Main theorem; dynamic maintenance of regression cost] Let $\varepsilon \in (0, 1/4)$ be an error parameter and $\mathbf{b}^{(1)}$ be the initial target label. Given $\varepsilon, \mathbf{A}, \mathbf{b}^{(1)}$, a stream of $T$ adaptively chosen, $K$-sparse updates to the label, Algorithm 4 outputs an estimate $\widehat{F}_i$ such that $\widehat{F}_i = (1 \pm \varepsilon)F(\mathbf{A}, \mathbf{b}^{(i)})$ for all $i$ with high probability.*

*Furthermore, the algorithm requires a preprocessing step in time $\widetilde{\mathcal{O}}\left(\mathrm{nnz}(\mathbf{A}) + \mathrm{poly}(d)\right)$. The amortized update time of the algorithm is*

$$\widetilde{\mathcal{O}}\left(\sqrt{K\,\mathrm{nnz}(\mathbf{A})}\left(\sqrt{\kappa(\mathbf{A})} + \varepsilon^{-3}\right)\right)$$

*per round.*

*Proof.* We focus on any fixed epoch of $T$ iterations. Let $\{\mathcal{A}_i\}_{i=1}^{\Gamma}$ be the collection of $\Gamma$ data structures maintained by the Algorithm 4 and $\mathcal{T}_i$ be the transcript between Algorithm 4 and the adversary at round $i$, consisting of the algorithm's output and the update requested by the adversary.

To handle a sequence of $T$ adaptive queries, consider the transcript $\mathcal{T}(R) = \{\mathcal{T}_1, \ldots, \mathcal{T}_T\}$, where $R$ denotes the internal randomness of Algorithm 4. Note that for a fixed iteration, $\mathcal{T}_i$ is $\left(\mathcal{O}\left(\frac{1}{\sqrt{T}\log(nT)}\right), 0\right)$-differentially private with respect to the algorithms $\mathcal{A}_1, \ldots, \mathcal{A}_\Gamma$, since the private median algorithm PrivMed is $\left(\mathcal{O}\left(\frac{1}{\sqrt{T}\log(nT)}\right), 0\right)$-differentially private. By the advanced composition of differential privacy, i.e., Theorem A.4, the transcript $\mathcal{T}$ is $\left(\mathcal{O}\left(1\right), \frac{1}{\mathrm{poly}(n)}\right)$-differentially private with respect to the algorithms $\mathcal{A}_1, \ldots, \mathcal{A}_\Gamma$.

Algorithm 4 runs $\Gamma$ instances of the data structure with error parameter $\varepsilon$. For any given round $i \in [T]$, we say that an instance $j \in [\Gamma]$ is correct if its output $f_{i,j}$ is within a $(1 \pm \varepsilon)$ factor of $F(\mathbf{A}, \mathbf{b}^{(i)})$ and incorrect otherwise. For a fixed $i$, let $Y_j$ be the indicator variable for whether $f_{i,j}$ is correct.

From the generalization properties of differential privacy, i.e., Theorem A.5, we have that for any fixed iteration $i$,

$$\mathbf{Pr}\left[\left|\frac{1}{\Gamma}\sum_{j\in[\Gamma]}Y_j - \mathbb{E}\left[Y\right]\right| \geq \frac{1}{10}\right] < \frac{1}{\mathrm{poly}(m,T)},$$

where $Y$ denotes the indicator random variable for whether a random instance of the algorithm $\mathcal{A}$ (not necessarily restricted to the $m$ instances maintained by the algorithm) is correct at the given round $i$. Since a random instance $\mathcal{A}$ has randomness that is independent of the adaptive update, then $\mathbb{E}\left[Y\right] \geq \frac{3}{4}$. Therefore, by a union bound over all $T$ rounds, we have

$$\mathbf{Pr}\left[\frac{1}{\Gamma}\sum_{i\in[\Gamma]}Y_i > 0.6\right] > 1 - \frac{1}{\mathrm{poly}(m,T)},$$

which implies that the output on the $i$th round is correct with probability at least $1 - \frac{1}{\mathrm{poly}(m,T)}$, since $T = d$. Then by a union bound over $i \in [T]$ for all $T$ rounds within an epoch, we have that the data structure answers all $T$ queries with probability $1 - \frac{1}{m^2}$, under the adaptively chosen updates. Finally,

by a union bound over all $m$ updates, we have that the algorithm succeeds with probability at least $1 - \frac{1}{m}$.

We now analyze the run-time of the algorithm. The preprocessing time follows from the guarantee of Lemma A.11. For update time, we amortize over each epoch. Within an epoch, we invoke $\Gamma = \mathcal{O}\left(\sqrt{T}\log(nT)\right)$ copies of the data structure in Section 3.2, and so we consider the preprocessing and update time from there and amortize over the epoch length $T$. By Lemma 3.1, each copy takes $\beta = \mathcal{O}\left(\sqrt{\kappa(\mathbf{A})}\,\text{nnz}(\mathbf{A})\log\frac{1}{\varepsilon} + \frac{\text{nnz}(\mathbf{A})}{\varepsilon^2}\log n + \frac{d}{\varepsilon^2}\log n\right)$ time to pre-process. For every step of update, each copy takes $\mathcal{O}\left(\frac{K}{\varepsilon^2}\log n\right)$ time by Lemma 3.2. Therefore, the amortized update time for every epoch of length $T = \mathcal{O}\left(\frac{\text{nnz}(\mathbf{A})}{\varepsilon^2 K}\right)$ is

$$\mathcal{O}\left(\frac{1}{T}\left(\Gamma\beta + \Gamma T\left(\frac{K}{\varepsilon^2}\log n\right)\right)\right) = \widetilde{\mathcal{O}}\left(\sqrt{K\,\text{nnz}(\mathbf{A})}\left(\sqrt{\kappa(\mathbf{A})} + \varepsilon^{-3}\right)\right).$$

This completes the proof. $\qquad\square$

### B.1    An Exact and Deterministic Algorithm

We now give a simple deterministic algorithm for the dynamic regression problem based on an SVD trick. Let $\mathbf{A} = \mathbf{U}\boldsymbol{\Sigma}\mathbf{V}^\top$ be the SVD of $\mathbf{A}$, where $\mathbf{U} \in \mathbb{R}^{n \times d}, \boldsymbol{\Sigma} \in \mathbb{R}^{d \times d}$ and $\mathbf{V} \in \mathbb{R}^{d \times d}$. The starting observation is that for any solution vector $\mathbf{x}$, we can write the regression cost as

$$\left\|\mathbf{A}\mathbf{x} - \mathbf{b}\right\| = \left\|\mathbf{U}\boldsymbol{\Sigma}\mathbf{V}^\top\mathbf{x} - \mathbf{b}\right\| = \left\|\boldsymbol{\Sigma}\mathbf{V}^\top\mathbf{x} - \mathbf{U}^\top\mathbf{b}\right\|, \tag{B.1}$$

since $\mathbf{U}$ is orthonormal. The goal is the maintain the solution vector $\mathbf{x} = \mathbf{A}^\dagger\mathbf{b}$ and the associated right-side quantity $\left\|\boldsymbol{\Sigma}\mathbf{V}^\top\mathbf{x} - \mathbf{U}^\top\mathbf{b}\right\|$.

Now suppose we compute $\mathbf{A}^\dagger \in \mathbb{R}^{d \times n}$ and $\mathbf{U}^\top \in \mathbb{R}^{d \times n}$ in pre-processing, and $\mathbf{A}^\dagger\mathbf{b}^{(1)}$ and $\mathbf{U}^\top\mathbf{b}^{(1)}$ in the first round. Then since all subsequent updates to $\mathbf{b}$ are all $K$-sparse, we only pay $\mathcal{O}(dK)$ time per step to maintain $\mathbf{A}^\dagger\mathbf{b}^{(i)}$ and $\mathbf{U}^\top\mathbf{b}^{(i)}$.

---

**Algorithm 5** A simple SVD-based algorithm for dynamic regression

---

**Input:** Design matrix $\mathbf{A} \in \mathbb{R}^{n \times d}$, its pseudoinverse $\mathbf{A}^\dagger \in \mathbb{R}^{d \times n}$ and its SVD $\mathbf{A} = \mathbf{U}\boldsymbol{\Sigma}\mathbf{V}^\top$, a sequence of labels $\mathbf{b}^{(i)} \in \mathbb{R}^n$
 1: Compute and store SVD $\mathbf{A} = \mathbf{U}\boldsymbol{\Sigma}\mathbf{V}^\top$, where $\mathbf{U} \in \mathbb{R}^{n \times d}, \boldsymbol{\Sigma} \in \mathbb{R}^{d \times d}, \mathbf{V} \in \mathbb{R}^{d \times d}$
 2: Compute and store $\mathbf{A}^\dagger$ from the SVD.    ▷In the 1st-round, compute and store $\mathbf{A}^\dagger\mathbf{b}^{(1)}, \mathbf{U}^\top\mathbf{b}^{(1)}$.
 3: **for** each update $\mathbf{b}^{(i)}$ **do**
 4:     Update and store $\mathbf{x}^{(i)} = \mathbf{A}^\dagger\mathbf{b}^{(i)}$
 5:     Update and store $\mathbf{U}^\top\mathbf{b}^{(i)}$
 6:     Output $F_i = \left\|\boldsymbol{\Sigma}\mathbf{V}^\top\mathbf{x}^{(i)} - \mathbf{U}^\top\mathbf{b}^{(i)}\right\|_2^2$

---

The algorithm is formally given by Algorithm 5. Observe that the algorithm always maintains the exact optimal regression cost. Moreover, the procedure does not require any randomness, and therefore it is adversarially robust to adaptive inputs. We formally claim the following guarantees of the algorithm.

**Theorem B.1** (Deterministic maintenance of regression costs)**.** *Given* $\mathbf{A}, \mathbf{b}^{(1)}$ *and a stream of adaptively chosen, $K$-sparse updates to the label, Algorithm 5 takes $\mathcal{O}(dK)$ time to update and maintain the exact regression cost $F(\mathbf{A}, \mathbf{b}^{(i)})$ at all iterations $i$. The pre-processing requires an SVD of $\mathbf{A}$, in $\mathcal{O}(n^2 d)$ time.*

## C    A Framework for Adversarial Robustness

In this section, we describe the benchmark framework that enables $Q$ adaptive queries to a data structure by using $\widetilde{\mathcal{O}}\left(\sqrt{Q}\right)$ copies of a non-adaptive data structure. The framework and corresponding analysis of correctness are simply compartmentalizations of the techniques in Hassidim et al. (2020);

Beimel et al. (2022); Attias et al. (2023). For the sake of completeness, we include them here and discuss additional applications. Namely, we show that through advanced composition of differential privacy, the private median of $\widetilde{\mathcal{O}}\left(\sqrt{Q}\right)$ copies protects the internal randomness of each non-adaptive data structure while still adding sufficiently small noise to guarantee accuracy. Moreover, we use amplification of privacy by sampling to only consider a small subset of the $\widetilde{\mathcal{O}}\left(\sqrt{Q}\right)$ non-adaptive data structures to further improve the runtime.

---

**Algorithm 6** Adaptive Algorithm Interaction

---

1: $r \leftarrow \mathcal{O}\left(\sqrt{Q}\log^2(nQ)\right), k \leftarrow \mathcal{O}\left(\log(nQ)\right)$
2: **for** $i \in [r]$ **do**
3:      Implement data structure $\mathcal{D}_i$ on the input
4: **for** each query $q_i$, $i \in [Q]$ **do**
5:      Let $S$ be a set of $k$ indices sampled (with replacement) from $[r]$
6:      For each $j \in [k]$, let $d_{i,j}$ be the output of $\mathcal{D}_{S_j}$ on query $q_i$
7:      $d_i \leftarrow \mathsf{PrivMed}(\{d_{i,j}\}_{j \in [k]})$, where $\mathsf{PrivMed}$ is $(1,0)$-DP

---

We first argue that Algorithm 6 maintains accuracy against $Q$ rounds of interaction with an adaptive adversary. Let $R = \{R^{(0)}, R^{(1)}, \ldots, R^{(r)}\}$, where $R^{(1)}, \ldots, R^{(r)}$ denotes the random strings used by the oblivious data structures $\mathcal{D}_1, \ldots, \mathcal{D}_r$ and $R^{(0)}$ denotes the additional randomness used by Algorithm 6, such as in the private median subroutine $\mathsf{PrivMed}$. Consider a transcript $T(R) = \{T_1, \ldots, T_Q\}$ such that for each $i \in [Q]$, we define $T_i = (q_i, d_i)$ to be the ordered pair consisting of the query $q_i$ and the corresponding answer $d_i$ by Algorithm 6 using the random string $R^{(0)}$, as well as the oblivious data structures $\mathcal{D}_1, \ldots, \mathcal{D}_r$ with random strings $R^{(1)}, \ldots, R^{(r)}$. We remark that $d_i$ is a random variable due to the randomness of each data structure, as well as the randomness of the private median subroutine $\mathsf{PrivMed}$.

We will first argue that the transcript $T_R$ is differentially private with respect to $R$. We emphasize that similar arguments were made in the streaming model by Hassidim et al. (2020) and in the dynamic model Beimel et al. (2022); Attias et al. (2023).

**Lemma C.1.** *For a fixed iteration, $T_i$ is $\left(\mathcal{O}\left(\frac{1}{\sqrt{Q}\log(nQ)}\right), 0\right)$-differentially private with respect to $R$.*

*Proof.* We first observe that $\mathsf{PrivMed}$ is $(1,0)$-differentially private on the outputs of the $r = \mathcal{O}\left(\sqrt{Q}\log^2(nQ)\right)$ data structures. Algorithm 6 samples $k = \mathcal{O}\left(\log(nQ)\right)$ groups of data structures from the $r$ total data structures. Thus by amplification via sampling, i.e., Theorem A.2, $\mathsf{PrivMed}$ is $\left(\mathcal{O}\left(\frac{1}{\sqrt{Q}\log(nQ)}\right), 0\right)$-differentially private. Therefore, $T_i$ is $\left(\mathcal{O}\left(\frac{1}{\sqrt{Q}\log(nQ)}\right), 0\right)$-differentially private with respect to $R$. $\qquad\square$

We next argue that the entire transcript is differentially private with respect to the randomness $R$.

**Lemma C.2.** *$T$ is $\left(\mathcal{O}\left(1\right), \frac{1}{\mathrm{poly}(nQ)}\right)$-differentially private with respect to $R$.*

*Proof.* By Lemma C.1, for each fixed iteration $i \in [Q]$, the transcript $T_i$ is $\left(\mathcal{O}\left(\frac{1}{\sqrt{Q}\log(nQ)}\right), 0\right)$-differentially private with respect to $R$. Note that the transcript $T$ is an adaptive composition of the transcripts $T_1, \ldots, T_Q$. Thus, by the advanced composition of differential privacy, i.e., Theorem A.4, the transcript $T$ is $\left(\mathcal{O}\left(1\right), \frac{1}{\mathrm{poly}(nQ)}\right)$-differentially private with respect to $R$. $\qquad\square$

We now prove the correctness of our unifying framework.

**Proof of Theorem 1.2:** For a fixed query $q_i$ with $i \in [Q]$, let $S$ be the corresponding set of $k$ indices sampled from $[r]$. Let $\mathcal{V}$ be the set of valid answers on query $q_i$. Let $I_j$ be an indicator variable

for whether the output $d_{i,j}$ on query $q_i$ by $\mathcal{D}_{S_j}$ is correct, so that $I_j = 1$ if $d_{i,j} \in \mathcal{V}$ and $I_j = 0$ if $d_{i,j} \notin \mathcal{V}$. By assumption, we have that for each $j \in [k]$,

$$\mathbf{Pr}\left[I_j = 1\right] \geq \frac{3}{4},$$

so that $\mathbb{E}\left[I_j\right] \geq \frac{3}{4}$. We define the random variable $I = \frac{1}{k}\sum_{j \in [k]} I_j$ so that by linearity of expectation, $\mathbb{E}\left[I\right] = \frac{1}{k}\sum_{j\in[k]} \mathbb{E}\left[I_j\right] \geq \frac{3}{4}$.

To handle a sequence of $Q$ adaptive queries, we consider the transcript $T(R) = \{T_1, \ldots, T_Q\}$ for the randomness $R = \{R^{(0)}, R^{(1)}, \ldots, R^{(r)}\}$ previously defined, i.e., for each $i \in [Q]$, $T_i = (q_i, d_i)$ is the ordered pair consisting of the query $q_i$ and the corresponding answer $d_i$ by Algorithm 6 using the random string $R^{(0)}$, as well as the oblivious data structures $\mathcal{D}_1, \ldots, \mathcal{D}_r$ with random strings $R^{(1)}, \ldots, R^{(r)}$. By Lemma C.2, we have that $T$ is $\left(\mathcal{O}\left(1\right), \frac{1}{\text{poly}(nQ)}\right)$-differentially private with respect to $R$.

For $j \in [k]$, we define the function $\text{success}(R^{(S_j)})$ to be the indicator variable for whether the output $d_{i,S_j}$ by data structure $D_{S_j}$ is successful on query $q_i$. For example, if $D$ is supposed to answer queries within $(1 + \alpha)$-approximation, then we define $\text{success}(R^{(S_j)})$ to be one if $d_{i,S_j}$ is within a $(1 + \alpha)$-approximation to the true answer on query $q_i$, and zero otherwise. From the generalization properties of differential privacy, i.e., Theorem A.5, we have

$$\mathbf{Pr}\left[\left|\frac{1}{k}\sum_{j \in [k]}\text{success}(R^{(S_j)}) - \mathbb{E}_{\overline{R}}\left[\text{success}(\overline{R})\right]\right| \geq \frac{1}{10}\right] < \frac{1}{\text{poly}(n, Q)},$$

for sufficiently small $\mathcal{O}\left(1\right)$. Therefore, by a union bound over all $Q$ queries, we have

$$\mathbf{Pr}\left[\frac{1}{k}\sum_{i \in [k]} I_i > 0.6\right] > 1 - \frac{1}{\text{poly}(n, Q)},$$

which implies that $d_i$ is correct on query $q_i$. Then by a union bound over $i \in [Q]$ for all $Q$ adaptive queries, we have that the data structure answers all $Q$ adaptive queries with high probability.    $\square$

## D    APPLICATIONS OF OUR FRAMEWORK

Theorem 1.2 has applications to a number of central problems in data science and machine learning. In this section, we formally describe the range queries, point queries, matrix-vector norm queries, and linear regression problems; we defer discussion of adaptive distance estimation, kernel density estimation, and nearest neighbor search to the the appendix.

### D.1    APPLICATION: MATRIX-VECTOR NORM QUERIES

In the matrix-vector norm query problem, we are given a matrix $\mathbf{A} \in \mathbb{R}^{n \times d}$ and we would like to handle $Q$ adaptive queries $\mathbf{x}^{(1)}, \ldots, \mathbf{x}^{(Q)}$ for an approximation parameter $\varepsilon > 0$ by outputting a $(1 + \varepsilon)$-approximation to $\|\mathbf{A}\mathbf{x}^{(i)}\|_p$ for each query $\mathbf{x}^{(i)} \in \mathbb{R}^d$ with $i \in [Q]$. Here we define $\|\mathbf{v}\|_p^p = \sum_{i \in [d]} |v_i|^p$ for a vector $\mathbf{v} \in \mathbb{R}^d$. Observe that computing $\mathbf{A}\mathbf{x}^{(i)}$ explicitly and then computing its $p$-norm requires $\mathcal{O}\left(nd\right)$ time. Thus for $n \gg d$, a much faster approach is to produce a subspace embedding, i.e., to compute a matrix $\mathbf{M} \in \mathbb{R}^{m \times d}$ with $m \ll n$, such that for all $\mathbf{x} \in \mathbb{R}^d$,

$$(1 - \varepsilon)\|\mathbf{A}\mathbf{x}\|_p \leq \|\mathbf{M}\mathbf{x}\|_p \leq (1 + \varepsilon)\|\mathbf{A}\mathbf{x}\|_p.$$

However, because subspace embeddings must be correct over all possible queries, the number of rows of $\mathbf{M}$ is usually $m = \Omega\left(\frac{d}{\varepsilon^2}\right)$ due to requiring correctness over an $\varepsilon$-net.

**Theorem D.1** (Indyk (2006); Li (2008))**.** *Given* $\mathbf{A} \in \mathbb{R}^{n \times d}$, $p \in (0, 2]$, *and an accuracy parameter* $\varepsilon > 0$, *there exists an algorithm that creates a data structure that uses* $\mathcal{O}\left(\frac{1}{\varepsilon^2}\log n\right)$ *bits of space and outputs a* $(1 + \varepsilon)$-*approximation to* $\|\mathbf{A}\mathbf{x}\|_p$ *for a query* $\mathbf{x} \in \mathbb{R}^d$, *with high probability, in time* $\mathcal{O}\left(\frac{d}{\varepsilon^2}\log n\right)$.

Theorem D.1 essentially creates a matrix $\mathbf{R} \in \mathbb{R}^{m \times n}$ of random variables sampled from $p$-stable distribution (Zolotarev, 1986) and then stores the matrix $\mathbf{RA}$. Once the query $\mathbf{x}$ arrives, the data structure then outputs a $(1 + \varepsilon)$-approximation to $\|\mathbf{Ax}\|_p$ by computing a predetermined function on $\mathbf{RAx}$. The restriction on $p \in (0, 2]$ is due to the fact that $p$-stable distributions only exist for $p \in (0, 2]$. From Theorem D.1 and Theorem 1.2, we have the following:

**Theorem D.2.** *Given $\mathbf{A} \in \mathbb{R}^{n \times d}$, $p \in (0, 2]$, and an accuracy parameter $\varepsilon > 0$, there exists an algorithm that creates a data structure that uses $\mathcal{O}\left(\frac{\sqrt{Q}}{\varepsilon^2} \log^2(nQ)\right)$ bits of space and outputs a $(1 + \varepsilon)$-approximation to $\|\mathbf{Ax}^{(i)}\|_p$ with $i \in [Q]$ for $Q$ adaptive queries $\mathbf{x}^{(1)}, \ldots, \mathbf{x}^{(Q)} \in \mathbb{R}^d$, with high probability, in time $\widetilde{\mathcal{O}}\left(\frac{d}{\varepsilon^2} \log^2(nQ) + \log^3(nQ)\right)$.*

## D.2 APPLICATION: LINEAR REGRESSION

In the linear regression problem, we are given a fixed matrix $\mathbf{A} \in \mathbb{R}^{n \times d}$ and we would like to handle $Q$ adaptive queries $\mathbf{b}^{(1)}, \ldots, \mathbf{b}^{(Q)}$, for an approximation parameter $\varepsilon > 0$, by outputting a $(1 + \varepsilon)$-approximation to $\min_{\mathbf{x} \in \mathbb{R}^d} \|\mathbf{Ax} - \mathbf{b}^{(i)}\|_2$ for each query $\mathbf{b}^{(i)} \in \mathbb{R}^n$ with $i \in [Q]$. For linear regression, we can again compute a subspace embedding $\mathbf{M} = \mathbf{SA} \in \mathbb{R}^{m \times n}$ and answer a query $\mathbf{b}^{(i)}$ by approximately solving $\min_{\mathbf{x} \in \mathbb{R}^d} \|\mathbf{SAx} - \mathbf{Sb}^{(i)}\|_2$, where $\mathbf{S}$ is a sketching matrix (Clarkson & Woodruff, 2013).

**Theorem D.3** (Clarkson & Woodruff (2013)). *Given $\mathbf{A} \in \mathbb{R}^{n \times d}$, $\mathbf{b} \in \mathbb{R}^n$, and an accuracy parameter $\varepsilon > 0$, there exists an algorithm that creates a data structure that uses $\mathcal{O}\left(\frac{d^2}{\varepsilon^2} \log^2(nQ)\right)$ bits of space and outputs a $(1 + \varepsilon)$-approximation to $\min_{\mathbf{x} \in \mathbb{R}^d} \|\mathbf{Ax} - \mathbf{b}\|_2$ with high probability.*

However, this may fail for multiple interactions with the data structure. For example, suppose the adversary learns the kernel of $\mathbf{S}$. Then the adversary could query some vector $\mathbf{b}^{(i)}$ in the kernel of $\mathbf{S}$ so that $\mathbf{Sb}^{(i)}$ is the all zeros vector, so that the output is the all zeros vector of dimension $d$, which could be arbitrarily bad compared to the actual minimizer. Thus the naïve approach is to maintain $Q$ subspace embeddings, one for each query, resulting in a data structure with space $\widetilde{\mathcal{O}}\left(\frac{Qd}{\varepsilon^2}\right)$. By comparison, Theorem D.3 and Theorem 1.2 yield the following:

**Theorem D.4.** *Given $\mathbf{A} \in \mathbb{R}^{n \times d}$ and an accuracy parameter $\varepsilon > 0$, there exists an algorithm that creates a data structure that uses $\mathcal{O}\left(\frac{\sqrt{Q}d^2}{\varepsilon^2} \log^3(nQ)\right)$ bits of space and with high probability, outputs $(1 + \varepsilon)$-approximations to $\min_{\mathbf{x} \in \mathbb{R}^d} \|\mathbf{Ax} - \mathbf{b}^{(i)}\|_2$ for $Q$ adaptive queries $\mathbf{b}^{(1)}, \ldots, \mathbf{b}^{(Q)}$.*

## D.3 APPLICATION: HALF-SPACE QUERIES

Given a set $P$ of $n$ points in $\mathbb{R}^d$, the range query or search problem asks us to pre-process $P$ so that given a region $R$, chosen from a predetermined family, one can quickly count or return the points in $P \cap R$. This is an extremely well-studied class of problems in computational geometry Toth et al. (2017) and the case where the regions $R$ are hyperplanes (also called half-spaces) is of special interest since many algebraic constraints can be "lifted" to be hyperplanes in a higher dimension.

Unfortunately, exact versions of the problem are known to have the "curse of dimensionality" and suffer from exponential dependence on $d$ in the query time (Brönnimann et al., 1993; Chazelle, 2000). Nonetheless, Chazelle et al. (2008) gave a data structure capable of answering hyperplane queries *approximately* with polynomial query time. Their notion of approximation is as follows: given a set of points $P$ in the unit $\ell_2$ ball, hyperplane $R$, and $\varepsilon > 0$, we return the number of points that are on a given side of the hyperplane $R$ up to additive error equal to the number of points in $P$ which lie within distance $\varepsilon$ of the boundary of $R$. We will refer to this query as an $\varepsilon$-approximate hyperplane query. Chazelle et al. (2008) proved the following theorem.

**Theorem D.5** (Chazelle et al. (2008)). *Given a set of points $P$ that lie in the unit $\ell_2$ ball, there exists a data structure that pre-processes $P$ using space $\widetilde{\mathcal{O}}\left(dn^{O(\varepsilon^{-2})}\right)$ such that any $\varepsilon$-approximate hyperplane range query is answered correctly with high probability. The query time is $\widetilde{\mathcal{O}}\left(d/\varepsilon^2\right)$.*

The data structure of Chazelle et al. (2008) is randomized and in particular employs randomized dimensionality reduction. Thus, it is feasible that queries might fail for multiple adaptive interactions

with the data structure. By utilizing our framework of Section C and Theorem 1.2, we can obtain the following robust guarantee.

**Theorem D.6.** *Given a set of points $P$ that lie in the unit $\ell_2$ ball, there exists a data structure which pre-processes $P$ using space $\widetilde{\mathcal{O}}\left(\sqrt{Q}dn^{O(\varepsilon^{-2})}\right)$ such that $Q$ adaptive $\varepsilon$-hyperplane range queries are answered correctly with high probability. The query time is $\widetilde{\mathcal{O}}\left(d/\varepsilon^2\right)$.*

### D.4 APPLICATION: POINT QUERIES ON TURNSTILE STREAMS

In the problem of point queries on turnstile streams, there exists a stream of $m$ updates. Each update specifies a coordinate $i \in [n]$ of an underlying frequency vector $f \in \mathbb{R}^n$ and changes $f_i$ by some amount between $\Delta_i \in [-\Delta, \Delta]$, where $\Delta = \text{poly}(n)$. Given any *constant* accuracy parameter $\varepsilon > 0$ any time $t \in [m]$, we define $f^{(t)}$ to be the frequency vector implicitly defined after the first $t$ updates. Then the point query problem is to output $f_i^{(t)}$ for various choices of $t \in [m]$ and $i \in [n]$ within an additive error of $\varepsilon \|f^{(t)}\|_1$.

**Theorem D.7** (Alman & Yu (2020)). *There exists an algorithm that uses space $\mathcal{O}\left(\log^2 n\right)$ bits, worst-case update time $\mathcal{O}\left(\log^{0.582} n\right)$, and query time $\mathcal{O}\left(\log^{1.582} n\right)$, that supports point queries with $\varepsilon = 0.1$ with high probability.*

An important quality of Theorem D.7 is that it significantly improves the update time over previous data structures, e.g., Charikar et al. (2004), at a cost in query time. By applying Theorem 1.2, we can avoid a blow-up in query time while still enjoying the update time improvements:

**Theorem D.8.** *There exists an algorithm that uses space $\mathcal{O}\left(\sqrt{Q}\log^3(nQ)\right)$ bits, has worst-case update time $\mathcal{O}\left(\sqrt{Q}\log^{1.582}(nQ)\right)$ and query time $\widetilde{\mathcal{O}}\left(\log^3(nQ)\right)$, and supports $Q$ adaptive point queries with $\varepsilon = 0.1$ and with high probability.*

## E ADAPTIVE DISTANCE ESTIMATION

In the adaptive distance estimation problem, there exists a set $X = \{\mathbf{x}^{(1)}, \ldots, \mathbf{x}^{(n)}\}$ of $n$ points in $\mathbb{R}^d$. Given an accuracy parameter $\varepsilon > 0$, the goal is to output a $(1 + \varepsilon)$-approximation to $\|\mathbf{x}^{(i)} - \mathbf{q}\|_p$ for each query $\mathbf{q}$ across all points $\mathbf{x}^{(i)} \in X$, while minimizing the space, query time, or pre-processing time for the corresponding data structures. The trivial solution stores all $n$ points and computes all $n$ distances to each query point and thus can handle an unlimited number of queries. Since each point has dimension $d$, the trivial solution uses space and query time $\mathcal{O}(nd)$. Cherapanamjeri & Nelson (2020) first improved the query time to $\widetilde{\mathcal{O}}\left(\frac{n+d}{\varepsilon^2}\right)$ at the cost of using $\widetilde{\mathcal{O}}\left(\frac{(n+d)d}{\varepsilon^2}\right)$ space and $\widetilde{\mathcal{O}}\left(\frac{nd^2}{\varepsilon^2}\right)$ pre-processing time. Like the trivial solution, the algorithm of Cherapanamjeri & Nelson (2020) also permits an arbitrary number of queries.

In this section, we first apply our framework to show a data structure that can handle $Q$ queries of approximate distances from a *specified point in $X$*, using query time $\widetilde{\mathcal{O}}\left(\frac{n+d}{\varepsilon^2}\right)$, pre-processing time $\widetilde{\mathcal{O}}\left(\frac{nd\sqrt{Q}}{\varepsilon^2}\right)$, and space $\widetilde{\mathcal{O}}\left(\frac{(n+d)\sqrt{Q}}{\varepsilon^2}\right)$. Hence for $d \gg n\sqrt{Q}$, our data structure already improves on the work of Cherapanamjeri & Nelson (2020).

However in this setting, each of the $Q$ queries returns only the approximate distance between a query point and a single point in $X$. By comparison, Cherapanamjeri & Nelson (2020) outputs approximate distances to all points in $X$ and moreover, follow-up work by Cherapanamjeri & Nelson (2022) improved the pre-processing time to $\widetilde{\mathcal{O}}\left(\frac{nd}{\varepsilon^2}\right)$. Therefore, we address these two shortcomings of our framework by giving a data structure that (1) handles the case where we return the approximate distances of all points in $X$ from $Q$ adaptive query points and (2) achieves pre-processing time $\widetilde{\mathcal{O}}\left(\frac{nd}{\varepsilon^2}\right)$.

For completeness, we now show correctness of our algorithm across all $Q$ adaptive queries, though we remark that the proof can simply be black-boxed into Theorem 1.2.

**Theorem E.1.** *With high probability, we have*

$$(1 - \varepsilon)\|\mathbf{x}_{i_q} - \mathbf{y}_q\|_2 \leq d_i \leq (1 + \varepsilon)\|\mathbf{x}_{i_q} - \mathbf{y}_q\|_2,$$

---

**Algorithm 7** Adaptive Distance Estimation

---
1: $r \leftarrow \mathcal{O}\left(\sqrt{Q}\log^2(nQ)\right), k \leftarrow \mathcal{O}\left(\log(nQ)\right)$
2: Let $\Pi_1, \ldots, \Pi_r \in \mathbb{R}^{m \times d}$ be a JL transformation matrix (see Theorem A.6 or Theorem A.7)
3: **for** $j \in [r]$ **do**
4:      Compute $\Pi_j \mathbf{x}_i$
5: **for** each query $(\mathbf{y}, i)$ with $\mathbf{y} \in \mathbb{R}^d, i \in [n]$ **do**            ▷Adaptive queries
6:      Let $S$ be a set of $k$ indices sampled (with replacement) from $[r]$
7:      **for** $j \in [k]$ **do**
8:          $d_{i,j} \leftarrow \|\Pi_{S_j}(\mathbf{x}_i - \mathbf{y})\|_2$
9:      $d_i \leftarrow \mathsf{PrivMed}(\{d_{i,j}\}_{j \in [k]})$, where PrivMed is $(1, 0)$-DP.
10:      **return** $d_i$

---

*for all $q \in [Q]$.*

*Proof.* Fix query $(\mathbf{y}_q, i_q)$ with $q \in [Q]$ and $i_q \in [n]$. Let $S$ be a set of $k$ indices sampled (with replacement) from $[r]$. By Theorem A.6 or Theorem A.7, then we have for each $j \in [k]$,

$$\mathbf{Pr}\left[(1 - \varepsilon)\|\mathbf{x}_{i_q} - \mathbf{y}_q\|_2 \le \|\Pi_{S_j}(\mathbf{x}_{i_q} - \mathbf{y}_q)\|_2 \le (1 + \varepsilon)\|\mathbf{x}_{i_q} - \mathbf{y}_q\|_2\right] \ge \frac{3}{4}.$$

Let $I_j$ be an indicator variable so that $I_j = 1$ if $(1 - \varepsilon)\|\mathbf{x}_{i_q} - \mathbf{y}_q\|_2 \le \|\Pi_{S_j}(\mathbf{x}_{i_q} - \mathbf{y}_q)\|_2 \le (1 + \varepsilon)\|\mathbf{x}_{i_q} - \mathbf{y}_q\|_2$ and $I_j = 0$ otherwise, so that we have $\mathbf{Pr}[I_j = 1] \ge \frac{3}{4}$, or equivalently, $\mathbb{E}[I_j] \ge \frac{3}{4}$. Let $I = \frac{1}{k}\sum_{j \in [k]} I_j$ so that by linearity of expectation, $\mathbb{E}[I] = \frac{1}{k}\sum_{j \in [k]} \mathbb{E}[I_j] \ge \frac{3}{4}$.

To address adaptive queries, we first note that PrivMed is $(1, 0)$-differentially private on the outputs of the $r$ Fast JL transforms. Since we sample $k = \mathcal{O}\left(\log(nQ)\right)$ groups from the $r = \mathcal{O}\left(\sqrt{Q}\log^2(nQ)\right)$ groups with replacement, then by amplification via sampling, i.e., Theorem A.2, PrivMed is $\left(\mathcal{O}\left(\frac{1}{\sqrt{Q}\log(nQ)}\right), 0\right)$-differentially private. Thus, by the advanced composition of differential privacy, i.e., Theorem A.4, the mechanism permits $Q$ adaptive queries and is $\left(\mathcal{O}(1), \frac{1}{\mathrm{poly}(nQ)}\right)$-differentially private. By the generalization properties of differential privacy, i.e., Theorem A.5, we have

$$\mathbf{Pr}\left[\left|\frac{1}{k}\sum_{j \in [k]} I_j - \mathbb{E}[I]\right| \ge \frac{1}{10}\right] < \frac{1}{\mathrm{poly}(Q, n)},$$

for sufficiently small $\mathcal{O}(1)$. Thus we have

$$\mathbf{Pr}\left[\frac{1}{k}\sum_{i \in [k]} I_i > 0.6\right] > 1 - \frac{1}{\mathrm{poly}(Q, n)},$$

which implies that $(1 - \varepsilon)\|\mathbf{x}_{i_q} - \mathbf{y}_q\|_2 \le d_i \le (1 + \varepsilon)\|\mathbf{x}_{i_q} - \mathbf{y}_q\|_2$. Therefore, by a union bound across $Q$ adaptive queries $(\mathbf{y}_q, \mathbf{x}_{i_q})$ with $q \in [Q]$, we have that $(1 - \varepsilon)\|\mathbf{x}_{i_q} - \mathbf{y}_q\|_2 \le d_i \le (1 + \varepsilon)\|\mathbf{x}_{i_q} - \mathbf{y}_q\|_2$ for all $q \in [Q]$ with high probability. $\qquad\square$

**Theorem E.2.** *There exists an algorithm that answers $Q$ adaptive distance estimation queries within a factor of $(1 + \varepsilon)$. For $\mathcal{O}\left(\left(\frac{\log d}{\varepsilon^2} + d\log d\right)\log(nQ)\right)$ query time, it stores $\mathcal{O}\left(\frac{n\sqrt{Q}\log^3(nQ)}{\varepsilon^2}\right)$ words of space. For $\mathcal{O}\left(\frac{d}{\varepsilon^2}\log(nQ)\right)$ query time, it stores $\mathcal{O}\left(\frac{n\sqrt{Q}\log^2(nQ)}{\varepsilon^2}\right)$ words of space.*

*Proof.* By Theorem A.7, each fast JL transform uses $\mathcal{O}\left(\frac{\log d}{\varepsilon^2} + d\log d\right)$ runtime and stores $m = \mathcal{O}\left(\frac{\log d}{\varepsilon^2}\right)$ rows. On the other hand, by Theorem A.6, each JL transform uses $\mathcal{O}\left(\frac{d}{\varepsilon^2} + d\log d\right)$ runtime and stores $m = \mathcal{O}\left(\frac{\log d}{\varepsilon^2}\right)$ rows. $\qquad\square$

By comparison, Cherapanamjeri & Nelson (2020) uses $\mathcal{O}\left(\frac{nd \log n}{\varepsilon^2}\right)$ words of space and $\mathcal{O}\left(\frac{d}{\varepsilon^2}\right)$ query time.

### E.1 FASTER PRE-PROCESSING TIME FOR ADAPTIVE DISTANCE ESTIMATION

In this section, we present an improved algorithm for Adaptive Distance Estimation, which allows the release of distances to *all* $n$ points in the dataset for a single query, matching the query time of Cherapanamjeri & Nelson (2020) with an improved space complexity of $O(\varepsilon^{-2}\sqrt{Q}n)$. Our results utilize a class of structured randomized linear transformations based on Hadamard matrices recursively defined below:

$$H_1 = [1] \qquad H_d = \begin{bmatrix} H_{d/2} & H_{d/2} \\ H_{d/2} & -H_{d/2} \end{bmatrix}.$$

The associated class of randomized linear transformations are now defined below:

$$\{D^j\}_{j\in[m]} \subset \mathbb{R}^{d\times d} \text{ s.t } D_{k,l}^j \overset{iid}{\sim} \begin{cases} \mathcal{N}(0, I) & \text{if } k = l \\ 0 & \text{otherwise} \end{cases}$$

$$\forall z \in \mathbb{R}^d : h(z) = \begin{bmatrix} H_d D^1 \\ H_d D^2 \\ \vdots \\ H_d D^m \end{bmatrix} \cdot z. \tag{SRHT}$$

Note that for any vector $z$, $h(z)$ may be computed in time $O(md \log d)$ due to the recursive definition of the Hadamard transform. We now let $\phi$ and $\Phi$ denote the pdf and cdf of a standard normal random variable, $\mathrm{Quant}_\alpha(\{a_i\}_{i\in[l]})$ the $\alpha^{th}$ quantile of a multi-set of real numbers $\{a_i\}_{i\in[l]}$ for any $l \in \mathbb{N}$ and define $\psi_r$ as follows:

$$\forall r > 0, a \in \mathbb{R} : \psi_r(a) := \min(|a|, r).$$

Through the remainder of the section, we condition on the event defined in the following lemma:

**Lemma E.3** (Claims 5.1 and 5.2 Cherapanamjeri & Nelson (2022)). *For any* $\delta \in \left(0, \frac{1}{2}\right)$, *with probability at least* $1 - \delta$:

$$\forall z \text{ s.t } \|z\| = 1 : 2 \leq \mathrm{Quant}_{\alpha-\beta/4}\left(\{h(z)_i\}_{i\in[md]}\right) \leq \mathrm{Quant}_{\alpha+\beta/4}\left(\{h(z)_i\}_{i\in[md]}\right) \leq 4$$

$$\forall z \text{ s.t } \|z\| = 1, r \geq 4\sqrt{\log(1/\varepsilon)} : \left(1 - \frac{\varepsilon}{2}\right) \leq \frac{1}{md} \cdot \sqrt{\frac{\pi}{2}} \cdot \sum_{i\in[md]} \psi_r(h_i(z)) \leq \left(1 + \frac{\varepsilon}{2}\right)$$

*as long as* $m \geq C\varepsilon^{-2} \log(2/\delta) \log^5(d/\varepsilon)$ *for some absolute constant* $C > 0$.

We will additionally require the following technical result from Cherapanamjeri & Nelson (2022), where for any vector $v \in \mathbb{R}^d$ and multiset $S = \{i_j\}_{j\in[k]}$ with $i_j \in [d]$, $v_S$ denotes the vector $[v_{i_1}, \ldots, v_{i_k}]$:

**Lemma E.4** (Theorem 1.4 Cherapanamjeri & Nelson (2022)). *Assume* $h : \mathbb{R}^d \to \mathbb{R}^{md}$ *(SRHT) satisfies the conclusion of Lemma E.3. Then, there is an algorithm,* $\mathrm{RetNorm}$, *which satisfies for all* $x \in \mathbb{R}^d$:

$$\mathbb{P}_S\left\{(1-\varepsilon) \cdot \|x\| \leq \mathrm{RetNorm}(h(x)_S) \leq (1+\varepsilon) \cdot \|x\|\right\} \geq 1-\delta \text{ for } S = \{i_j\}_{j\in[k]} \text{ with } i_j \overset{iid}{\sim} \mathrm{Unif}([md])$$

*when* $k \geq C\varepsilon^{-2} \log(2/\varepsilon) \log(2/\delta)$ *for some* $C > 0$. *Furthermore,* $\mathrm{RetNorm}$ *runs in time* $O(k)$.

With these primitives, we will construct our data structure for adaptive distance estimation. Our constructions is formally described in Algorithm 8.

---

**Algorithm 8** Adaptive Distance Estimation with SRHTs

---

1: $m \leftarrow C\varepsilon^{-2}\log^6(2dn/\varepsilon)$
2: Let $h$ be an SRHT as defined in SRHT                            ▷Revealed to analyst
3: $r \leftarrow C\sqrt{Q}\log^3(nd), k \leftarrow C\varepsilon^{-2}\log(2/\varepsilon)\log(2nd)$
4: **for** $i \in [n]$ **do**
5:     Compute $y_i = h(x_i)$
6:     **for** $j \in [r]$ **do**
7:         Let $S_{i,j}$ be a set of $k$ indices sampled with replacement from $[md]$
8: $l \leftarrow C\log(nd)$
9: **for** $j \in 1 : Q$ **do**                                         ▷Adaptive queries
10:     Receive query $q_j$
11:     $v_j \leftarrow h(q_j)$
12:     **for** $i \in [n]$ **do**
13:         Let $\{t_{i,j,p}\}_{p\in[l]}$ be a set of $l$ indices sampled (with replacement) from $[r]$
14:         **for** $p \in [l]$ **do**
15:             $d_{i,j,p} \leftarrow \text{RetNorm}((v_j - y_i)_{S_{i,t_{i,j,p}}})$
16:         $d_{i,j} \leftarrow \text{PrivMed}(\{d_{i,j,p}\}_{p\in[l]})$, where PrivMed is $(\mathcal{O}(1), 0)$-DP.
17:     **return** $\{d_{i,j}\}_{i\in[n]}$

---

The proof of correctness of Algorithm 8 will follow along similar lines to that of Algorithm 6 with a more refined analysis of the privacy loss incurred due to the adaptivity of the data analyst. In particular, each input query results in $n$ different queries made to a differentially private mechanism PrivMed leading to a total of $nQ$ queries. A naïve application of Theorem 1.2 would thus result in a data structure with space complexity scaling as $\widetilde{O}(n^{3/2}\sqrt{Q})$ as opposed to the desired $\widetilde{O}(n\sqrt{Q})$ and query complexity $\widetilde{O}(\varepsilon^{-2}nd)$. The key insight yielding the improved result is the privacy loss incurred by a single query is effectively amortized across $n$ independent differentially private algorithms each capable of answering $Q$ adaptively chosen queries correctly with high probability.

To start, we first condition on the event in Lemma E.3 and assume public access to the correspondingly defined SRHT $h$. We now use $R$ to denote the randomness used to instantiate the multisets, $S_{i,j}$, in Algorithm 8 and decompose it as follows $R = \{R_i\}_{i\in[n]}$ with $R_i = \{R_{i,j}\}_{j\in[r]}$ where $R_{i,j}$ corresponds to the randomness used to generate the set $S_{i,j}$ and the random elements $t_{i,p}$. As in the proof of Theorem 1.2, we define a transcript $T = \{T_j\}_{j\in[Q]}$ with $T_j = (q_j, \{d_{i,j}\}_{i\in[n]})$ denoting the $j^{th}$ query and the responses returned by Algorithm 8 as a single transaction.

**Lemma E.5.** *For all $i \in [n], j \in [Q]$, $T_j$ is $\left(o\left(\frac{1}{\sqrt{Q}\log(nQ)}\right), 0\right)$-differentially private with $R_i$.*

*Proof.* The proof is identical to that of Lemma C.1 with the observation that each transaction $T_j$ only results in a single query to a differentially private mechanism operating on $R_i$. □

**Lemma E.6.** *For all $i \in [n]$, $T$ is $\left(o(1), \frac{1}{\text{poly}(nQ)}\right)$-differentially private with respect to $R_i$.*

*Proof.* The proof is identical to Lemma C.2 and follows from Theorem A.4 and Lemma E.5. □

We now prove the correctness of our improved procedure for adaptive distance estimation.

**Proof of Theorem 1.3:** We condition on the event in the conclusion of Lemma E.3 start by bounding the failure probability of a single query. The bound for the whole sequence of adaptively chosen queries follows by a union bound. Now, fixing $i \in [n]$ and $j \in [Q]$, note that the sub-transcript $T^{(j)} = \{T_p\}_{p\in[j-1]}$ is $\left(o(1), \frac{1}{\text{poly}(nQ)}\right)$-differentially private with respect to $R_i$. Furthermore, define the indicator random variables:

$$\forall p \in [l] : W_p := \mathbf{1}\left\{(1-\varepsilon) \cdot \|q_j - x_i\| \leq \text{RetNorm}\left((v_j - y_i)_{S_{i,t_{i,j,p}}}\right) \leq (1+\varepsilon) \cdot \|q_j - x_i\|\right\}$$

Additionally, defining $W := \sum_{p=1}^{l} W_p$, we get by the differential privacy of the sub-transcript, $T^{(j)}$, Lemma E.4 and Theorem A.5:

$$\mathbb{P}\left\{W \leq \frac{3}{4} \cdot l\right\} \leq \frac{1}{400 \cdot (nQ)^2}.$$

Consequently, we get from Theorem A.3 and another union bound:

$$\mathbb{P}\left\{(1-\varepsilon) \cdot \|q_j - x_i\| \leq d_{i,j} \leq (1+\varepsilon) \cdot \|q_j - x_i\|\right\} \geq 1 - \frac{1}{200 \cdot (nQ)^2}.$$

A subsequent union bound over all $i \in [n], j \in [Q]$ yields:

$$\mathbb{P}\left\{\forall i \in [n], j \in [Q] : (1-\varepsilon) \cdot \|q_j - x_i\| \leq d_{i,j} \leq (1+\varepsilon) \cdot \|q_j - x_i\|\right\} \geq 1 - \frac{1}{200 \cdot (nQ)}.$$

A final union bound over the conclusion of Lemma E.3 concludes the proof. The runtime guarantees follow from the fact that for all $z \in \mathbb{R}^d$, $h(z)$ is computable in time $O(md \log d)$ and the runtime guarantees of RetNorm. $\qquad\square$

## F  ADAPTIVE KERNEL DENSITY ESTIMATION

Kernel density estimation is an important problem in learning theory and statistics that has recently attracted significant interest, e.g., (Charikar & Siminelakis, 2017; Backurs et al., 2018; Charikar et al., 2020; Bakshi et al., 2022). In the adaptive kernel density estimation problem, the input is a set $X = \{\mathbf{x}^{(1)}, \ldots, \mathbf{x}^{(n)}\}$ of $n$ points in $\mathbb{R}^d$. Given an accuracy parameter $\varepsilon > 0$ and a threshold parameter $\tau > 0$, the goal is to output a $(1+\varepsilon)$-approximation to the quantity $\frac{1}{n} \sum_{i \in [n]} k(\mathbf{x}^{(i)}, \mathbf{q})$, for a kernel function $k$ under the promise that the output is at least $\tau$. A standard approach is to sample $\mathcal{O}\left(\frac{1}{\tau \varepsilon^2}\right)$ points and then use $\mathcal{O}\left(\frac{d}{\tau \varepsilon^2}\right)$ query time to output the empirical kernel density for a specific query. Backurs et al. (2019) give an algorithm for kernel density estimation that uses $\mathcal{O}\left(\frac{1}{\tau \varepsilon^2}\right)$ space and $\mathcal{O}\left(\frac{d}{\sqrt{\tau} \varepsilon^2}\right)$ query time, improving over the standard sampling approach.

**Theorem F.1.** *Backurs et al. (2019) Given $\varepsilon, \tau > 0$, there exists a data structure $D$ that uses $\mathcal{O}\left(\frac{1}{\tau \varepsilon^2}\right)$ space and $\mathcal{O}\left(\frac{d}{\varepsilon^2 \sqrt{\tau}}\right)$ query time that outputs a $(1+\varepsilon)$-approximation $D(\mathbf{y})$ to a kernel density estimation query $\mathbf{y}$ that has value at least $\tau$, i.e.,*

$$\mathbf{Pr}\left[|D(\mathbf{y}) - \mathrm{KDE}(X, \mathbf{y})| \leq \varepsilon \cdot \mathrm{KDE}(X, \mathbf{y})\right] \geq \frac{3}{4}.$$

However, the analysis for both these algorithms fails for the adaptive setting, where there can be dependencies between the query and the data structure. By using the data structure of Backurs et al. (2019) as a subroutine, our framework immediately implies an algorithm for adaptive kernel density estimation that uses $\widetilde{\mathcal{O}}\left(\frac{\sqrt{Q}}{\tau \varepsilon^2}\right)$ space and $\mathcal{O}\left(\frac{d \log Q}{\sqrt{\tau} \varepsilon^2}\right)$ query time to answer each of $Q$ adaptive queries.

---

**Algorithm 9** Adaptive Kernel Density Estimation

---

**Input:** Number $Q$ of queries, accuracy $\varepsilon$, threshold $\tau$
1: $r \leftarrow \mathcal{O}\left(\sqrt{Q} \log^2 Q\right)$
2: **for** $i \in [r]$ **do**                    ▷Pre-processing
3:   Let $T_i$ be a KDE data structure
4: **for** each query $\mathbf{y}_q \in \mathbb{R}^d$ with $q \in [Q]$ **do**           ▷Adaptive queries
5:   Let $S$ be a set of $k$ indices sampled (with replacement) from $[r]$
6:   **for** $i \in [k]$ **do**
7:     Let $D_i$ be the output of $T_{S_i}$ on query $\mathbf{y}_q$
8:   **return** $d_q = \mathsf{PrivMed}(\{D_i\}_{i \in [k]})$, where $\mathsf{PrivMed}$ is $(1, 0)$-DP.

---

For completeness, we now show adversarial robustness of our algorithm across $Q$ adaptive queries. Again we remark that the proof can simply be black-boxed into Theorem 1.2, though we include the specific kernel density details in the following proof as a warm-up for the following section.

**Lemma F.2.** *Algorithm 9 answers $Q$ adaptive kernel density estimation queries within a factor of $(1 + \varepsilon)$, provided each query has value at least $\tau$.*

*Proof.* Fix query $\mathbf{y}_q \in \mathbb{R}^d$ with $q \in [Q]$. Let $S$ be a set of $k$ indices sampled (with replacement) from $[r]$. Then by Theorem F.1, we have that for each $j \in [k]$,

$$\mathbf{Pr}\left[\left|D_{S_j}(\mathbf{y}) - \mathrm{KDE}(X, \mathbf{y})\right| \leq \varepsilon \cdot \mathrm{KDE}(X, \mathbf{y})\right] \geq \frac{3}{4}.$$

Let $I_j$ be an indicator variable so that $I_j = 1$ if $\left|D_{S_j}(\mathbf{y}) - \mathrm{KDE}(X, \mathbf{y})\right| \leq \varepsilon \cdot \mathrm{KDE}(X, \mathbf{y})$ and $I_j = 0$ otherwise, so that we have $\mathbf{Pr}[I_j = 1] \geq \frac{3}{4}$ or equivalently, $\mathbb{E}[I_j] \geq \frac{3}{4}$. Let $I = \frac{1}{k}\sum_{j\in[k]} I_j$ so that $\mathbb{E}[I] = \frac{1}{k}\sum_{j\in[k]} \mathbb{E}[I_j] \geq \frac{3}{4}$.

To handle adaptive queries, we first note that PrivMed is $(1, 0)$-differentially private on the outputs of the $r$ kernel density estimation data structures. We sample $k = \mathcal{O}(\log Q)$ indices from the $r = \mathcal{O}\left(\sqrt{Q}\log^2 Q\right)$ data structures with replacement. Thus by amplification via sampling, i.e., Theorem A.2, PrivMed is $\left(\mathcal{O}\left(\frac{1}{\sqrt{Q}\log Q}\right), 0\right)$-differentially private. By the advanced composition of differential privacy, i.e., Theorem A.4, our algorithm can answer $Q$ adaptive queries with $\left(\mathcal{O}(1), \frac{1}{\mathrm{poly}(Q)}\right)$-differentially privacy. By the generalization properties of differential privacy, i.e., Theorem A.5, we have

$$\mathbf{Pr}\left[\left|\frac{1}{k}\sum_{j\in[k]} I_j - \mathbb{E}[I]\right| \geq \frac{1}{10}\right] < 0.01,$$

for sufficiently small constant $\mathcal{O}(1)$ in the private median algorithm PrivMed. Therefore,

$$\mathbf{Pr}\left[\frac{1}{k}\sum_{i\in[k]} I_i > 0.6\right] > 0.99,$$

so that $|d_q - \mathrm{KDE}(X, \mathbf{y}_q)| \leq \varepsilon \cdot \mathrm{KDE}(X, \mathbf{y})$ across $Q$ queries $\mathbf{y}_q$ with $q \in [Q]$. $\qquad\square$

**Theorem F.3.** *There exists an algorithm that uses $\mathcal{O}\left(\frac{\sqrt{Q}\log^2 Q}{\tau\varepsilon^2}\right)$ space and answers $Q$ adaptive kernel density estimation queries within a factor of $(1 + \varepsilon)$, provided each query has value at least $\tau$. Each query uses $\mathcal{O}\left(\frac{d\log(nQ)}{\varepsilon^2\sqrt{\tau}}\right)$ runtime.*

By comparison, random sampling, e.g., Charikar & Siminelakis (2017), uses $\frac{Q}{\tau\varepsilon^2}$ samples to answer $Q$ queries and each query uses $\mathcal{O}\left(\frac{d}{\tau\varepsilon^2}\right)$ runtime and using $Q$ copies of the data structure by Backurs et al. (2019) uses $\mathcal{O}\left(\frac{Q}{\tau\varepsilon^2}\right)$ space and $\mathcal{O}\left(\frac{d}{\varepsilon^2\sqrt{\tau}}\right)$ runtime.

## F.1 UNLIMITED ADAPTIVE QUERIES FOR KERNEL DENSITY ESTIMATION

In this section, we go beyond the limits of our framework and analyze the case where there may be an unbounded number of adversarial queries.

**Theorem 1.4.** *Suppose the kernel function $k$ is $L$-Lipschitz in the second variable for some $L > 0$, i.e., $|k(\mathbf{x}, \mathbf{y}) - k(\mathbf{x}, \mathbf{z})| \leq L\|\mathbf{y} - \mathbf{z}\|_2$ for all $\mathbf{x}, \mathbf{y}, \mathbf{z} \in \mathbb{R}^d$. Moreover, suppose that for all $\|\mathbf{x} - \mathbf{y}\|_2 \leq \rho$, we have $k(\mathbf{x}, \mathbf{y}) \leq \frac{\tau}{3}$. Then an algorithm that produces a kernel density estimation data structure $D$ that is $L$-Lipschitz over a set $X$ of points with diameter at most $\Delta$ and outputs a $(1+\varepsilon)$-approximation to KDE queries with value at least $\tau$ with probability at least $1 - \delta$ using space $S(n, \varepsilon, \tau, \log \delta)$ and query time $T(n, \varepsilon, \tau, \log \delta)$, then there exists a KDE data structure that with probability at least $0.99$, outputs a $(1 + \varepsilon)$-approximation to any number of KDE queries with value at least $\tau$ using space $S\left(n, \mathcal{O}(\varepsilon), \mathcal{O}(\tau), \mathcal{O}\left(d\log\frac{(\Delta+\rho)L}{\varepsilon\tau}\right)\right)$ and query time $T\left(n, \mathcal{O}(\varepsilon), \mathcal{O}(\tau), \mathcal{O}\left(d\log\frac{(\Delta+\rho)L}{\varepsilon\tau}\right)\right)$.*

*Proof.* Given a set $X \subseteq \mathbb{R}^d$ of $n$ points with diameter $\Delta$, let $\mathcal{N}$ be an $\frac{\varepsilon\tau}{L}$-net over a ball of radius $\Delta + \rho$ that contains $X$. More formally, let $B$ be a ball of radius $(\Delta + \rho)$ that contains $X$ and for

every $\mathbf{y} \in B$, there exists a point $\mathbf{z} \in \mathcal{N}$ such that $\|\mathbf{y} - \mathbf{z}\|_2 \leq \frac{\varepsilon\tau}{L}$. We can construct the net greedily so that $|\mathcal{N}| \leq \left(\frac{2(\Delta + \rho)L}{\varepsilon\tau}\right)^d$.

We implement a data structure $D$ that answers each (non-adaptive) kernel density estimation query with multiplicative approximation $\left(1 + \frac{\varepsilon}{3}\right)$ for any kernel density estimation query with value at least $\frac{\tau}{2}$, with probability at least $1 - \delta$, where $\delta \leq \frac{1}{100|\mathcal{N}|}$. Then by a union bound, $D$ correctly answers each kernel density estimation query in $\mathcal{N}$ with probability at least $0.99$.

Let $\mathbf{q} \in \mathbb{R}^d$ be an arbitrary query such that $\mathrm{KDE}(X, \mathbf{q}) \geq \tau$. By assumption, we have that $\|\mathbf{q} - \mathbf{x}\|_2 \leq \rho$ for some $\mathbf{x} \in X$ and thus $\mathbf{q} \in B$. By the definition of $\mathcal{N}$, there exists some $\mathbf{y} \in \mathcal{N}$ such that $\|\mathbf{q} - \mathbf{y}\|_2 \leq \frac{\varepsilon\tau}{3L}$. Then since $k$ is $L$-Lipschitz in the second variable, we have

$$|\mathrm{KDE}(X, \mathbf{q}) - \mathrm{KDE}(X, \mathbf{y})| = \left| \frac{1}{n} \sum_{\mathbf{x} \in X} k(\mathbf{x}, \mathbf{q}) - \frac{1}{n} \sum_{\mathbf{x} \in X} k(\mathbf{x}, \mathbf{y}) \right| \leq \frac{L}{n} \|\mathbf{q} - \mathbf{y}\|_2 \leq \frac{\varepsilon\tau}{3n}.$$

Hence, $\mathrm{KDE}(X, \mathbf{q}) \geq \tau$ implies that $\mathrm{KDE}(X, \mathbf{y}) \geq \frac{\tau}{2}$. Let $K_\mathbf{y}$ be the output of the data structure $D$ on query $\mathbf{y}$. Then by correctness of $D$ on $\mathcal{N}$ for any query with threshold at least $\frac{\tau}{2}$, we have

$$|K_\mathbf{y} - \mathrm{KDE}(X, \mathbf{y})| \leq \frac{\varepsilon}{3} \mathrm{KDE}(X, \mathbf{y}).$$

Let $K_\mathbf{q}$ be the output of the data structure $D$ on query $\mathbf{y}$. Since the algorithm itself is $L$-Lipschitz, then

$$|K_\mathbf{q} - K_\mathbf{y}| \leq L\|\mathbf{q} - \mathbf{y}\|_2 \leq \frac{\varepsilon\tau}{3}.$$

Therefore by the triangle inequality, we have that

$$|K_\mathbf{q} - \mathrm{KDE}(X, \mathbf{q})| \leq |K_\mathbf{q} - K_\mathbf{y}| - |K_\mathbf{y} - \mathrm{KDE}(X, \mathbf{y})| - |\mathrm{KDE}(X, \mathbf{y}) - \mathrm{KDE}(X, \mathbf{q})|$$
$$\leq \frac{\varepsilon\tau}{3} + \frac{\varepsilon}{3} \mathrm{KDE}(X, \mathbf{y}) + \frac{\varepsilon\tau}{3n}.$$

Since $\mathrm{KDE}(X, \mathbf{y}) \leq \mathrm{KDE}(X, \mathbf{q}) + \frac{\varepsilon\tau}{3n}$, then it follows that

$$|K_\mathbf{q} - \mathrm{KDE}(X, \mathbf{q})| \leq \frac{\varepsilon\tau}{3} + \frac{\varepsilon}{3} \mathrm{KDE}(X, \mathbf{q}) + \frac{\varepsilon^2\tau}{n} + \frac{\varepsilon\tau}{3n} \leq \varepsilon \, \mathrm{KDE}(X, \mathbf{q}),$$

for $n \geq 6$. $\qquad\square$

In particular, sampling-based algorithms for kernels that are Lipschitz are also Lipschitz. Thus to apply Theorem 1.4, it suffices to identify kernels that are $L$-Lipschitz and use the data structure of Theorem F.1. To that end, we note that the kernels $k(\mathbf{x}, \mathbf{y}) = \frac{C}{C + \|\mathbf{x} - \mathbf{y}\|_2}$ for $C > 0$ and $k(\mathbf{x}, \mathbf{y}) = Ce^{-\|\mathbf{x} - \mathbf{y}\|_2}$ are both Lipschitz for some function of $C$. In particular, we have

$$|k(\mathbf{x}, \mathbf{y}) - k(\mathbf{x}, \mathbf{z})| = \left| \frac{C}{C + \|\mathbf{x} - \mathbf{y}\|_2} - \frac{C}{C + \|\mathbf{x} - \mathbf{z}\|_2} \right|$$
$$= \frac{C|\|\mathbf{x} - \mathbf{z}\|_2 - \|\mathbf{x} - \mathbf{y}\|_2|}{(C + \|\mathbf{x} - \mathbf{y}\|_2)(C + \|\mathbf{x} - \mathbf{z}\|_2)}$$
$$\leq \frac{\|\mathbf{y} - \mathbf{z}\|_2}{C},$$

so $k(\mathbf{x}, \mathbf{y}) = \frac{C}{C + \|\mathbf{x} - \mathbf{y}\|_2}$ is $\frac{1}{C}$-Lipschitz. Similarly, since $e^{-x}$ is 1-Lipschitz, then

$$|k(\mathbf{x}, \mathbf{y}) - k(\mathbf{x}, \mathbf{z})| = Ce^{-\|\mathbf{x} - \mathbf{y}\|_2} - Ce^{-\|\mathbf{x} - \mathbf{z}\|_2}$$
$$\leq C|\|\mathbf{x} - \mathbf{z}\|_2 - \|\mathbf{x} - \mathbf{y}\|_2| \leq C\|\mathbf{y} - \mathbf{z}\|_2,$$

so $k(\mathbf{x}, \mathbf{y}) = Ce^{-\|\mathbf{x} - \mathbf{y}\|_2}$ is $C$-Lipschitz.

# G    EMPIRICAL EVALUATION

We empirically demonstrate the space and query time efficiency of our framework of Section C. We consider the problem of $\ell_2$ norm estimation where queries $q_1, q_2, \ldots$ are generated in an *adaptive* fashion and our goal is to output an estimate of $\|q_i\|_2$ for all $i$. This setting is a special case of adaptive distance estimation and captures the essence of our adversarial robustness framework. In addition, this same setting was investigated empirically in prior works Cherapanamjeri & Nelson (2020).

**Experimental Setup.**    Consider the setting of Algorithm 6: it creates $r$ copies of an underlying randomized data structure and upon a query, it subsamples $k$ of them and outputs an answer aggregated via the private median. In our setting, the underlying algorithm will be the *fast Johnson-Lindenstrauss* (JL) transform which is defined as follows: it is the matrix $PHD : \mathbb{R}^d \to \mathbb{R}^m$ where $D$ is a diagonal matrix with uniformly random $\pm 1$ entries, $H$ is the Hadamard transform, and $P$ is a sampling matrix uniformly samples $m$ rows of $HD$. Our algorithm will initialize $r$ copies of this matrix where the sampling matrix $P$ and diagonal $D$ will be the randomness which is "hidden" from the adversary. Upon query $q$, we sample $k$ different Fast JL data structures, input $q$ to all of them, and proceed as in Algorithm 6. Note that this setting exactly mimics the theoretical guarantees of Section C and is exactly Algorithm 7 of Section E. In our experiments, $d = 4096, m = 250, r = 200$, and $k = 5$. These are exactly the parameters chosen in prior works Cherapanamjeri & Nelson (2020). We will have 5000 adaptive queries $q_i$ which are described shortly. Our experiments are done on a 2021 M1 Macbook Pro with 32 gigabytes of RAM. We implemented all algorithms in Python 3.5 using Numpy. The Hadamard transform code is from Andoni et al. (2015)[1] and we use Google's differential privacy library[2] for the private median implementation.

**Baselines.**    We will consider three baselines. **JL** will denote a standard (Gaussian) JL map from dimension 4096 to 250. **Baseline 1** will denote the algorithm of Cherapanamjeri & Nelson (2020). At a high level, it instantiates many independent copies of the standard Gaussian JL map and only feeds an incoming query into a select number of subsampled data structures. Note that our experimental setting is mimicking exactly that of Cherapanamjeri & Nelson (2020) where the same parameters $r$ (number of different underlying data structures) and $k$ (number of subsampled data structures to use for a query) were used. This ensures that both our algorithm and theirs have access to the same number of *different* JL maps and thus allows us to compare the two approaches on an equal footing. The last baseline, denoted as **Baseline 2**, is the main algorithm of Cherapanamjeri & Nelson (2022) which is the optimized version of Cherapanamjeri & Nelson (2020). At a high level, their algorithm proceeds similarly to that of Cherapanamjeri & Nelson (2020), except they employ Hadamard transforms (after multiplying the query entry-wise by random Gaussians), rather than using Gaussian JL maps. Furthermore, instead of subsampling, their algorithm feeds an incoming query into all the different copies of the Hadamard transform, and subsamples the coordinates of the concatenated output for norm estimation. We again set the parameters of their algorithm to match that of our algorithm and **Baseline 1** by using $r$ copies of their Hadamard transform and subsampling $mk$ total coordinates. We refer to the respective papers for full details of their algorithms.

**Summary of adaptive queries.**    Our input queries are the same adaptive queries used in Cherapanamjeri & Nelson (2020). To summarize, let $\Pi$ denote the map used in the **JL** benchmark stated above. The $i$-th query for $1 \le i \le 5000$ will be of the form $q_i = \sum_{j=1}^{i} (-1)^{W_i} z_i$, which we then normalize to have unit norm. The $z_i$ are standard Gaussian vectors. $W_i$ is the indicator variable for the event $\|\Pi(z_i - e_1)\|_2 \le \|\Pi(z_i + e_1)\|_2$ where $e_1$ is the first standard basis vector. Intuitively, the queries become increasingly correlated with the matrix $\Pi$ since we successively "augment" the queries in a biased fashion. See Section 5 of Cherapanamjeri & Nelson (2020) for a more detailed discussion of the adaptive inputs.

**Results.**    Our results are shown in Figure 1. In Figure 1a, we plot the norm estimated by each of the algorithms in each of the queries across iterations. We see that the naïve **JL** map increasingly deviates from the true value of $1.0$. This is intuitive as the adaptive queries are increasingly correlated

---

[1]available in https://github.com/FALCONN-LIB/FFHT
[2]available in https://github.com/google/differential-privacy

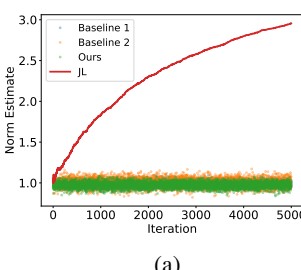 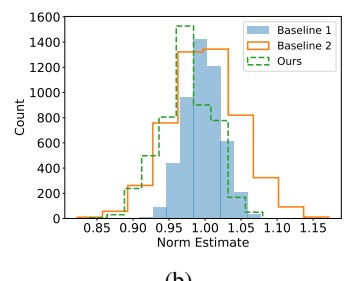 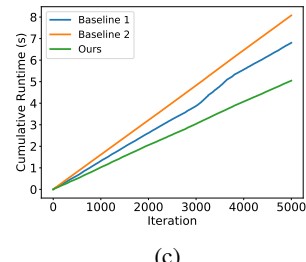

(a)                    (b)                    (c)

Fig. 1: Figures for our experiments.

with the map $\Pi$. The performance of all other algorithms are indistinguishable in Figure 1a. Thus, we only zoom into the performances of our algorithm and **Baseline 1** and **Baseline 2**, shown in Figure 1b. For these three algorithms, we plot a histogram of answers outputted by the respective algorithms across all iterations. We see that the algorithm of Cherapanamjeri & Nelson (2020), shown in the blue shaded histogram, is the most accurate as it has the smallest deviations from the true answer of $1.0$. Our algorithm, shown in green, is noisier than **Baseline 1** since it has a wider range of variability. This may be due to the fact that we use a differentially private median algorithm, which naturally incurs additional noise. Lastly, **Baseline 2** is also noisier than **Baseline 1** and comparable to our algorithm. This may be due to the fact that the algorithm of Cherapanamjeri & Nelson (2022) requires very fine-tuned constants in their theoretical bounds, which naturally deviate in practice. Lastly, Figure 1c shows the cumulative runtime of all three algorithms across all iterations. Our algorithm, shown in green, is the fastest while **Baseline 2** is the slowest. This is explained by the fact that **Baseline 2** calculates many more Hadamard transforms than our algorithm does.

