# OpenReview forum: "Robust Algorithms on Adaptive Inputs from Bounded Adversaries"
_ICLR.cc/2023/Conference — ICLR 2023 poster_

### Official Review · Reviewer_iYm7 · 2022-10-24

**Confidence:** 3
**Correctness:** 4
**Technical Novelty And Significance:** 3
**Empirical Novelty And Significance:** Not applicable
**Recommendation:** 8

**Clarity, Quality, Novelty And Reproducibility:**

The paper is clear, interesting and to my understanding novel. There are no experiments so reproducibility is irrelevant.

**Strength And Weaknesses:**

Strengths:
- interesting topic
- well written paper
- neat way of incorporating differential privacy result

Weaknesses:
- no empirical evaluation. It would be interesting to see how the algorithms perform in practice against other algorithms that are not-robust to adversarial intervention.

**Summary Of The Paper:**

The paper studies algorithms that are robust to adversaries of limited power. This is interesting since the input to an algorithm in many situations can change dynamically, either natural or as the consequence of  an attack. The paper considers two such models: in the first one the adversary can only alter a few elements of the input (and this model is applied to linear algebraic algorithms). In the second setting the adversary is allowed a limited number of (adaptive) queries to the algorithm. In the latter setting, an idea from differential privacy is employed to hide the internals of the algorithm to the adversary and thus guarantee robustness. This setting is applied to a sequence of problems.



**Summary Of The Review:**

This is a good paper and I recommend acceptance.

---

> ### Author Response · Authors · 2022-11-08
> **Response of Reviewer iYm7**
>
> > no empirical evaluation. It would be interesting to see how the algorithms perform in practice against other algorithms that are not-robust to adversarial intervention.
>
> For the sake of completeness, we remark that we perform empirical evaluations in Appendix A of the full version included in the supplementary material at the time of the submission. Our results in Figure 1 show that as a simple proof-of-concept against three common baselines, our algorithm performs similarly if not better in accuracy, but also runs significantly faster.

---

> > ### Comment · Reviewer_iYm7 · 2022-11-25
> > **Thanks**
> >
> > Thanks for your response and for pointing me to the empirical evaluation.

---

### Official Review · Reviewer_Kvip · 2022-10-25

**Confidence:** 2
**Correctness:** 3
**Technical Novelty And Significance:** 3
**Empirical Novelty And Significance:** Not applicable
**Recommendation:** 6

**Clarity, Quality, Novelty And Reproducibility:**

The robustness in estimation is in high demand even for the sequential estimation problem. The sequential estimation problem under adversarial queries is thus well-motivated. For the first problem of the sequential least-square objective value estimation, the authors employ L0 restriction for an adversary. The L0 restriction is a reasonable restriction on the adversary and is motivated to investigate. The algorithm for the first problem builds from a novel combination of the JL sketch technique and differentially private median estimation. The theoretical analyses provide an interesting characterization of the amortized update time and preprocessing time.

One this unclear to me in the first problem is the implication of Theorem 2.1. The authors state that a naive approach costs $\mathrm{nnz}(\mathbf{A})+\mathrm{poly}(d)$ for the amortized time. On the other hand, the amortized time for the proposed algorithm is $\sqrt{K\mathrm{nnz}(\mathbf{A})}/\epsilon^3$. So, if $\epsilon \le (\sqrt{K\mathrm{nnz}(\mathbf{A})}/(\mathrm{nnz}(\mathbf{A})+\mathrm{poly}(d)))^{1/3}$, is it better to use the naive approach? If so, it is better to clarify such a threshold for $\epsilon$ is meaningfully small.

I'm currently unsure about the significance of the results for the second problem of the general sequential estimation under adversarial queries. The unclarity comes from a lack of detailed comparison with Beimel et al.'s results. Beimel et al. also develop a general reduction framework for the sequential estimation problem under adversarial queries. The basic approach looks equivalent; they also utilize the differentially private median estimation and amplification via sampling technique to guarantee robustness against malicious queries. Similar theoretical results are presented in Theorem 3.1 in the original paper. Hence, the detailed comparison with Beimel et al.'s results is mandatory for clarifying the significance of this paper's results regarding the general sequential estimation problem.


Minor comment:
- While the amortized time is $\tilde{O}(1/\epsilon^{2.5})$ in Theorem 1.1, it is $\tilde{O}(1/\epsilon^{3})$ in Theorem 2.1. I guess Theorem 2.1 is correct.



**Strength And Weaknesses:**

Strength:
- Sequential estimation under adversarial queries is interesting and well-motivated.
- This paper gives novel and significant results for the sequential least-square objective value estimation under L0-restricted adversarial queries.

Weakness:
- The implication of the sequential least-square objective value estimation results is somewhat unclear.
- This paper lacks a detailed comparison with some crucial related works, which makes me unsure about the significance of the results.

**Summary Of The Paper:**

The authors investigate the space complexity and amortized round time or query time of the sequential (approximate) query answering problem under adversarial queries. First, they deal with the problem of sequentially answering the least-square objective value with an adversary who modifies the outcome values with L0 restriction. Second, they develop a general framework that can convert an algorithm working with stochastic queries to one for adversarial queries. They demonstrate many applications of their framework, including matrix-vector norm, linear regression, half-space query, point query on turnstile streams, distance estimation, and kernel density estimation. A general idea for dealing with these tasks is to hide the internal state from the adversary by utilizing the differentially private median.

**Summary Of The Review:**

The sequential estimation problem under adversarial queries is interesting and crucial. The first result for the sequential estimation of the least-square objective value is novel and significant. I'm thus currently recommending acceptance. However, I have some concerns regarding the implication of the first result and the originality of the second result of the general framework for adversarial sequential estimation. I'd like the authors to address such concerns in the rebuttal.

---

> ### Author Response · Authors · 2022-11-08
> **Response to Reviewer Kvip**
>
> > The implication of the sequential least-square objective value estimation results is somewhat unclear.
>
> Estimating the least-square objective value is important in applications such as distributed functional monitoring, c.f., [CMY11], where a number of sites are continuously monitored by a central coordinator. In this model, the coordinator can choose to perform a certain action if the regression cost becomes too high or too low. For example, if the cost is too high then perhaps the current set of features needs to be expanded to obtain better prediction, while if the cost is low enough, perhaps the coordinator is satisfied with the current predictor. However, these sites can be sophisticated devices, such as sensors, computers, or even entire networks. Therefore, certain sites may act in particular ways depending on the actions of the coordinator and certain sites may even act maliciously. Thus it is important for the algorithm to be adversarially robust.
>
> We will add this discussion as motivation for the problem in the introduction in the full version of the paper.
>
> [CMY11] Graham Cormode, S. Muthukrishnan, Ke Yi: Algorithms for distributed functional monitoring. ACM Trans. Algorithms 7(2): 21:1-21:20 (2011)
>
> > One this unclear to me in the first problem is the implication of Theorem 2.1. The authors state that a naive approach costs for the amortized time $\text{nnz}(\mathbf{A})+\text{poly}(d)$. On the other hand, the amortized time for the proposed algorithm is $\sqrt{K\text{nnz}(\mathbf{A})}/\epsilon^3$. So, if $\epsilon\le(\sqrt{K\text{nnz}(\mathbf{A})}/\(\text{nnz}(\mathbf{A})+\text{poly}(d)))^{1/3}$, is it better to use the naive approach? If so, it is better to clarify such a threshold for is meaningfully small.
>
> Indeed, it is better to use the naive approach for such a regime of $\epsilon$ though (1) it is not uncommon for approximation algorithms to perform worse than the na\"{i}ve algorithm once the accuracy parameter is sufficiently small and (2) our algorithm is still correct for this regime of $\epsilon$.
>
> We also remark that in Appendix B.4, we also give a simple deterministic algorithm with runtime $O(dK)$. Our DP approach improves upon that and gives $O(\sqrt{K\text{nnz}(\mathbf{A})}) = O(d\sqrt{K})$ when $\mathbf{A}$ is relatively square.
>
> > This paper lacks a detailed comparison with some crucial related works, which makes me unsure about the significance of the results.
> > I'm currently unsure about the significance of the results for the second problem of the general sequential estimation under adversarial queries. The unclarity comes from a lack of detailed comparison with Beimel et al.'s results. Beimel et al. also develop a general reduction framework for the sequential estimation problem under adversarial queries. The basic approach looks equivalent; they also utilize the differentially private median estimation and amplification via sampling technique to guarantee robustness against malicious queries. Similar theoretical results are presented in Theorem 3.1 in the original paper. Hence, the detailed comparison with Beimel et al.'s results is mandatory for clarifying the significance of this paper's results regarding the general sequential estimation problem.
>
> Indeed, the basic techniques of differentially private median estimation and amplification via sampling are similar. However, since Beimel et al. focuses on graph problems in the dynamic model, their main goal is to optimize the update time of their algorithms restricted to evolving and adaptive inputs but static queries. On the other hand, we consider general problems in the centralized setting, thus achieving applications that permit both adaptive updates and adaptive queries. For example, our applications to matrix-vector norm queries, linear regression, half-space queries, and point queries on turnstile streams all permit adaptive queries from a large domain.
>
> We also emphasize that for several applications of our framework such as adaptive distance estimation or adaptive kernel density estimation, we additionally use specific sophisticated techniques for these problems to further improve our algorithmic guarantees. As a simple example, for adaptive kernel density estimation, we provide a data structure robust to an arbitrary number of adaptive queries, which cannot be handled by the techniques of Beimel et. al.
>
> > While the amortized time is $\tilde{O}(1/\epsilon^{2.5})$ in Theorem 1.1, it is $\tilde{O}(1/\epsilon^{3})$ in Theorem 2.1. I guess Theorem 2.1 is correct.
>
> Yes, Theorem 2.1 is correct. We have fixed this typo, thanks for pointing it out!
>
> In short, we believe that your concerns will be fully resolved with a number of light editorial changes in the updated full version of the paper. Thus if you find our response satisfactory, we hope you will consider raising your score. Otherwise, we would be happy to engage in further discussions to clarify any misconceptions we may have about your concerns.

---

### Official Review · Reviewer_n2fk · 2022-10-25

**Confidence:** 1
**Correctness:** 2
**Technical Novelty And Significance:** 2
**Empirical Novelty And Significance:** Not applicable
**Recommendation:** 6

**Clarity, Quality, Novelty And Reproducibility:**

Clarity: being unable to follow all the details, I find the presentation of the paper actually quite clear.
Quality: I cannot evaluate the quality due to lack of the knowledge of this specific field.
Reproducibility: it is a purely theoretical work.

**Strength And Weaknesses:**

The work seems theoretically solid. However, it doesn't seem to fit into the representation learning context, especially the robustness questions. First, the models are primarily linear ones and the authors didn't give insight how these might scale to non-linear ones. Second, the perturbation in the target is interesting but the authors didn't explain what kind of applications it may find in reality. The paper seems fairly theoretical and very difficult to follow without the specific background. I'm wondering if the work would fit better in another venue.

**Summary Of The Paper:**

This paper introduces training algorithms that are robust to bounded dynamics in the data, with a focus on label updates.


----------------------
Update: I thank the authors for their informative feedback. I have updated my overall recommendation but remain unconfident about it.

**Summary Of The Review:**

My main concern is that it may not fit into the ICLR context. I'll be happy to change my evaluation if the authors and other reviewers may enlighten me regarding this point.

---

> ### Author Response · Authors · 2022-11-08
> **Response to Reviewer n2fk**
>
> > The work seems theoretically solid. However, it doesn't seem to fit into the representation learning context, especially the robustness questions. First, the models are primarily linear ones and the authors didn't give insight how these might scale to non-linear ones. Second, the perturbation in the target is interesting but the authors didn't explain what kind of applications it may find in reality. The paper seems fairly theoretical and very difficult to follow without the specific background. I'm wondering if the work would fit better in another venue.
> > My main concern is that it may not fit into the ICLR context. I'll be happy to change my evaluation if the authors and other reviewers may enlighten me regarding this point.
>
> Adversarial machine learning is a central point of focus in current research. In particular, there has recently been a large emphasis on theoretical guarantees for adversarially robust algorithms at top machine learning venues, e.g., consider the following works in the last two years listed below.
>
> Furthermore, adversarially robust algorithms are one of the main use cases of differential privacy, a well established study in machine learning in its own right. We add to the growing body of knowledge listed in below which strengthens the connection between robust algorithms and differential privacy.
>
> Kristian Georgiev, Samuel B. Hopkins: Privacy Induces Robustness: Information-Computation Gaps and Sparse Mean Estimation. NeurIPS 2022
>
> Edith Cohen, Xin Lyu, Jelani Nelson, Tamas Sarlos, Moshe Shechner, Uri Stemmer: On the Robustness of CountSketch to Adaptive Inputs. ICML 2022: 4112-4140
>
> Alexandr Andoni, Daniel Beaglehole: Learning to Hash Robustly, Guaranteed. ICML 2022: 599-618
>
> Vladimir Braverman, Avinatan Hassidim, Yossi Matias, Mariano Schain, Sandeep Silwal, Samson Zhou: Adversarial Robustness of Streaming Algorithms through Importance Sampling. NeurIPS 2021: 3544-3557
>
> Avinatan Hassidim, Haim Kaplan, Yishay Mansour, Yossi Matias, Uri Stemmer: Adversarially Robust Streaming Algorithms via Differential Privacy. NeurIPS 2020
>
> Yeshwanth Cherapanamjeri, Jelani Nelson: On Adaptive Distance Estimation. NeurIPS 2020
>
> > it is a purely theoretical work.
>
> We remark that we perform empirical evaluations in Appendix A of the full version included in the supplementary material at the time of the submission. Our results in Figure 1 show that as a simple proof-of-concept against three common baselines, our algorithm performs similarly if not better in accuracy, but also runs significantly faster.
>
> In summary, we emphasize that not only does our paper complement our theoretical results with empirical evaluations, but also its contents makes progress on an active topic in the machine learning community and thus would be a great fit for ICLR. We thus hope that these points will alleviate your concerns and that you will consider raising your score. Otherwise, we would also be happy to clarify any misunderstandings we may have about your concerns.

---

### Official Review · Reviewer_w9hf · 2022-10-27

**Confidence:** 3
**Clarity, Quality, Novelty And Reproducibility:** Good.
**Correctness:** 4
**Technical Novelty And Significance:** 3
**Empirical Novelty And Significance:** Not applicable
**Recommendation:** 8

**Strength And Weaknesses:**

I am not quite convinced by the first problem. I think the problem of how to not resolve linear regression under updates is natural, but I am not sure why you would only care about approximating the squared error rather computing a near-optimal solution. Typically, the loss is a means to the end of finding a good $x$, not the end in itself.

I found the second result much more interesting and general. The use of differential privacy is nice (and is somewhat reminiscent of the adaptive data analysis paper of Dwork et. al). It gives them  black box results for distance and kernel density estimation, which they are able to improve on in some settings. Similar techniques seem to have been used earlier in work on the streaming model.

**Summary Of The Paper:**

This paper considers the setting where an algorithm/data structure has to give responses to a sequence of adversarially chosen inputs.
The paper considers tow separate settings:
1. The consider linear regression $min_x \|Ax =b\|_2^2$ where the target $b$ can be updated in $K$ locations at each step. The goal is approximate the squared error.
2. They consider a setting where you have a non-adaptive data structure for a problem. They show that in order to answer $Q$ adversarial queries,  rather than use $Q$ independent copies of the. data structure, one can use privacy amplification techniques to only maintain $\sqrt{Q}$ copies.

**Summary Of The Review:**

I think this paper is nice and ought to be accpted.

---

> ### Author Response · Authors · 2022-11-08
> **Response to Reviewer w9hf**
>
> > I am not quite convinced by the first problem. I think the problem of how to not resolve linear regression under updates is natural, but I am not sure why you would only care about approximating the squared error rather computing a near-optimal solution. Typically, the loss is a means to the end of finding a good $x$, not the end in itself.
>
> Approximating the squared error loss is important in applications such as distributed functional monitoring, c.f., [CMY11], where a number of sites are continuously monitored by a central coordinator, who can choose to perform a certain action if the regression cost becomes too high or too low. For example, if the cost is too high then perhaps the current set of features needs to be expanded to obtain better prediction, while if the cost is low enough, perhaps the coordinator is satisfied with the current predictor. On the other hand, these sites can be sensors, computers, or even entire networks and so certain sites may act in particular ways depending on the actions of the central coordinator. Certain sites may even act maliciously and thus it is important for the algorithm to be adversarially robust.
>
> Thanks for the question -- we will add this discussion as explicit motivation for the problem.
>
> [CMY11] Graham Cormode, S. Muthukrishnan, Ke Yi: Algorithms for distributed functional monitoring. ACM Trans. Algorithms 7(2): 21:1-21:20 (2011)
>
> > Similar techniques seem to have been used earlier in work on the streaming model.
>
> We remark that although the idea of using differential privacy to protect the internal randomness of algorithms has been previously used in the streaming model, we use the amplification of privacy by sampling to further improve the query time of our general framework.
> Moreover, we emphasize for several applications of our framework such as adaptive distance estimation or adaptive kernel density estimation, we use specific techniques catered to these problems to further improve our algorithmic guarantees.

---

### Author Response · Authors · 2022-11-08
**Thanks to all reviewers**

We thank all reviewers for their valuable feedback. In particular, we greatly appreciate encouraging comments such as:
- I found the second result...interesting and general. The use of differential privacy is nice (and is somewhat reminiscent of the adaptive data analysis paper of Dwork et. al). It gives them black box results for distance and kernel density estimation, which they are able to improve on in some settings. (Reviewer w9hf)
- The work seems theoretically solid. (Reviewer n2fk)
- I find the presentation of the paper actually quite clear. (Reviewer n2fk)
- Sequential estimation under adversarial queries is interesting and well-motivated. (Reviewer Kvip)
- This paper gives novel and significant results for the sequential least-square objective value estimation under L0-restricted adversarial queries. (Reviewer Kvip)
- The algorithm for the first problem builds from a novel combination of the JL sketch technique and differentially private median estimation. The theoretical analyses provide an interesting characterization of the amortized update time and preprocessing time. (Reviewer Kvip)
- The sequential estimation problem under adversarial queries is interesting and crucial. The first result for the sequential estimation of the least-square objective value is novel and significant. (Reviewer Kvip)
- interesting topic, well written paper, neat way of incorporating differential privacy result (Reviewer iYm7)
- The paper is clear, interesting and to my understanding novel (Reviewer iYm7)

We provide our responses to specific questions of each reviewer below. We hope they resolve all initial comments and concerns raised by the reviewers and we would be happy to answer any remaining questions!

---

### Author Response · Authors · 2022-11-16
**Update check**

Hi everyone,

We have uploaded updated versions of the extended abstract and the full version of the paper that incorporates the initial reviewer feedback. For the sake of clarity, we have highlighted large additional text changes in blue in the full version of the paper.

As the discussion period is drawing to a close, we wanted to check whether the reviewers or the area chair had any additional questions, comments, or concerns that we could potentially address, particularly in relation to (but not limited to) our responses to the initial reviews.

Thanks again for your consideration!

---

### Decision · Program_Chairs · 2023-01-20

**Decision:**

Accept: poster

**Justification For Why Not Higher Score:**

In my view, this is a very technical paper that might be interesting only to a small subset of the community. However, I am fine if the senior AC decides to bump it up.

**Justification For Why Not Lower Score:**

Clear accept.

**Metareview: Summary, Strengths And Weaknesses:**

This submission studies algorithms robust to adaptive/adversarial input generated from sources with bounded capabilities.

All the reviewers found the paper interesting and solid. I recommend acceptance.

**Note From Pc:**

if the above contains the word "oral" or "spotlight" please see: "oral" presentation means -> notable-top-5% and "spotlight" means -> notable-top-25%. As stated in our emails, we are disassociating presentation type from AC recommendations